



# Copulas for hydroclimatic applications - A practical note on common misconceptions and pitfalls

Faranak Tootoonchi[1], Jan Olaf Haerter[2], Olle Räty[3], Thomas Grabs[1], Mojtaba Sadegh[4], Claudia Teutschbein[1]

[1] Department of Earth Sciences, Uppsala University, Uppsala, Sweden
[2] Niels Bohr Institute, University of Copenhagen, Copenhagen, Denmark
[3] Finnish Meteorological Institute, Helsinki, Finland
[4] Department of Civil Engineering, Boise State University, Boise, USA

Correspondence to: Faranak Tootoonchi (faranak.tootoonchi@geo.uu.se)

**Abstract.** For most hydroclimatic applications, precipitation and temperature are of particular interest as they strongly affect the water cycle, can easily be measured and are often readily available from many meteorological stations worldwide. To account for precipitation and temperature variability, their co-dependence and their correlation, several multivariate analysis methods have been adopted in the hydroclimatic literature in recent years. In line with the steadily rising number of publications on this topic, the notion of copula-based probability distribution has also attracted tremendous interest to deal with the
complexity of compound events in the multidimensional context. A copula is a function that connects a multivariate distribution to its one-dimensional margins, which allows for a joint distribution of random variables with great flexibility for the marginal distribution. However, there seems to be a lack of comprehensive understanding of the fundamental requirements of the copula concept such as the strength and significance of correlation between variables, autocorrelation effects and the choice of representative copula families, which potentially compromises the robustness of projections of future environmental
processes and natural hazards. Therefore, by combining a systematic literature review with a specific hydroclimatic case study in Sweden, we illustrate a practical approach to copula-based modeling, which (1) provides end-users with an overview of necessary requirements, statistical assumptions and consequential limitations of copulas, (2) highlights possible pitfalls and misconceptions, and (3) offers a decision support framework for the application of copulas to support researchers and decision makers in addressing climatological hazards and sustainable development, thereby demystifying what is currently an area of
great confusion.

## 1. Introduction

Water touches every aspect of our lives, from public health to safety, to the foundation of our economy. Alterations to the hydroclimatic drivers of the water cycle form a potential threat, as they influence socioeconomic, ecological and climate systems (Vogel et al., 2015). In this paper, we adopt the definition of hydroclimatology as the "study of the influence of climate
upon the waters of the land" originally proposed by Walter Langbein in the 1970's and later further expanded to include hydrometeorology (i.e., the land-atmosphere interface) as well as the surface and near-surface water processes of evaporation, runoff, groundwater recharge and interception (Wendland, 1987). As such, hydroclimatology encompasses a large number of





applications, e.g. (1) the assessment of a changing climate on water resources (including water quantity and quality), (2) the analysis of feedback of the water cycle variations on climate change, and (3) the evaluation of anthropogenic impacts such as

land-use change on local climate conditions and water resources.

For most hydroclimatic applications, precipitation and temperature are of main interest for modeling and future projections as both variables can easily be measured and are typically readily available from many meteorological stations worldwide (see e.g. NOAA Climate Data Online) or from global gridded databases such as NCEP/NCAR (Kalnay et al., 1996), ERA-interim (Dee et al., 2011) or JRA-55 reanalysis (Ebita et al., 2011). Both variables also strongly affect the water cycle: precipitation

as a direct component of the water cycle, and temperature as a proxy for energy availability, which in turn is of key interest for estimating evapotranspiration as well as snow accumulation and melt, both of which exert influence on the amount and timing of runoff. As a result, the spatiotemporal trends of precipitation and temperature greatly affect environmental processes, control natural hazards such as floods and droughts, and influence water quality and availability (Alidoost et al., 2019; Ribeiro et al., 2019). Therefore, future socio-economic development and sustainable use of water resources is highly dependent on

reliable representations and modeling of these two variables (Alexandrov and Hoogenboom, 2000; Phiri et al., 2019; Satgé et al., 2019).

On top of the natural variability of these meteorological variables, anthropogenic climate change has resulted in more surface water being transferred to the atmosphere, increasing the saturation water vapor mixing ratio there (Ashraf Vaghefi et al., 2014). These processes have exacerbated droughts (Trenberth et al., 2014) and intensified rainfall and flooding events

(Mangini et al., 2018). To account for precipitation and temperature variability, numerous studies have focused on assessing these two variables separately at different spatiotemporal scales (Hundecha and Bárdossy, 2005; Maftei et al., 2011; Teutschbein and Seibert, 2012). However, such hydroclimatic variables are interconnected and have a combined impact on the water balance. Focusing on these variables separately ignores their co-dependence, which may lead to an insufficient characterization of their variability and co-dependence (Hao et al., 2018). Hence, to consider the co-dependence and correlation

between these two variables, there have been attempts to study them simultaneously in a compound context. For example, Benestad and Haugen (2007) examined the joint probabilities of high temperatures and heavy precipitation during spring in Norway to evaluate flood hazards. Piani and Haerter (2012) studied compound precipitation and temperature data derived from different stations in Germany to assess and correct biases in climate model data by using their joint distribution. Zscheischler, Orth, and Seneviratne (2017) used bivariate return periods of precipitation and temperature over Europe to investigate crop

yield variability. Boeck et al. (2011) explored the combined influence of precipitation and temperature on plant communities and concluded that the negative effects of a dry spell combined with a heatwave on plant growth and functioning were considerably greater than the sum of their individual impacts.

All these studies highlight the importance of jointly considering the compound effects of precipitation and temperature dynamics in hydroclimatic studies. However, an accurate assessment of the joint distribution of these two variables – especially

in the context of a changing climate - is not straightforward, as their degree of co-dependence can vary with changes in spatiotemporal scales and for extreme events (Li et al. 2014). For example, Trenberth et al. (2003) argue that at large spatial



scales the intensity of extreme precipitation events should increase exponentially with temperature. However, because of the influence of local atmospheric circulations at local scales, precipitation and temperature are likely to be weakly correlated in many regions (Chou and Neelin, 2004; Colacino and Dell'osso, 1978; Dai and Bloecker, 2019). The strength of precipitation-

temperature correlation is additionally affected by the temporal resolution under consideration. At the annual resolution AghaKouchak et al. (2014) found a significant negative correlation between anomalies of temperature and precipitation over California. At the monthly resolution, Trenberth (2018) found a strong negative correlation between temperature and precipitation in the mid-latitude continental regions during summer. However at the daily resolution there is typically weak statistical evidence of a link between temperature and precipitation in many regions (Ferrari et al., 2018).

Despite these limitations at relatively fine spatio-temporal resolutions, several multivariate analysis methods have been adopted in the hydroclimatic literature in recent years (Fig. 1(a)). A keyword search in Scopus (Burnham, 2006), resulted in 914 published papers since 1970 until the 1st of June 2020 when combining the terms '*multivariate*' and '*hydrology*', and 6,090 publications when combining the terms '*multivariate*' and '*climate*'. It should be noted that the total number of papers published per year has nearly doubled since 2010 (Fig. 1(a)). These papers utilize a number of different multivariate methods,

including, for instance, multivariate analysis of variance (Shukla and Gedam, 2019), multivariate regression (Abdulelah Al-Sudani et al., 2019) or principal component analysis (Wang et al. 2019).

In line with the increasing use of multivariate methods, the application of copula-based probability distributions - a specific type of multivariate approaches - has also attracted tremendous interest (Fig. 1(b)) to deal with the complexity of compound events in the multidimensional space. The copula concept is particularly advantageous, because it allows for constructing a

joint distribution of random variables without any constraint on their marginal distributions (Fig. 2) (AghaKouchak et al., 2010). A copula is a function that connects a multivariate distribution to its one-dimensional margins (Schweizer, 1991). It is based on the early works of Sklar (1959) and has been widely used in different fields of study such as in finance (Cherubini et al., 2013) or in the energy sector (Ghoddusi, 2017). Since the early 2000's, copula methods have been adopted in hydrological modeling, which was triggered by the study of Salvadori and De Michele (2010). Since then researchers have applied copulas

to study interactions between a variety of different hydroclimatic variables related to e.g. sediment transport (Shojaeezadeh et al., 2020), flood hazard (Didier et al., 2019) heatwave mortality (Mazdiyasni et al., 2017) or to study hydrology and climate in general (Fig. 1(b)).

Copulas have been used in both theoretical and applied analyses of hydroclimatic data. Theoretical studies adopted the copula-based methodology for instance to gain a better understanding of the dependence structures of temperature and precipitation,

in different parts of the world (Cong and Brady, 2012; Lazoglou and Anagnostopoulou, 2019). Applied studies adopted the copula-based methodology to examine e.g. agricultural droughts (Wang et al. 2019; Ribeiro et al. 2019) or joint effects of temperature and precipitation extremes on vegetation growth (Alidoost et al., 2019; Cong and Brady, 2012). In addition, numerous climate-change impact studies applied copulas to bias-correct future projections by global or regional climate models (Piani and Haerter 2012; Li et al. 2014; Gennaretti, Sangelantoni, and Grenier 2015; Vrac and Friederichs 2015; Rana,





Moradkhani, and Qin 2017; Chen et al. 2018; Li and Babovic 2019; Mesbahzadeh et al. 2019) even though the usefulness of multivariate bias-correction methods for hydrological purposes has recently been put into question (Räty et al., 2018).

Despite the growing number of applied studies that adopt copulas, practitioners have to rely on statistical literature, which often focuses on the theoretical concept behind copulas (e.g., Genest, Rémillard, and Beaudoin 2009 or Berg 2009) while only a few papers also relate to practical challenges encountered in hydroclimatic research (e.g., Dupuis 2007). Hence, there seems

to be a lack of comprehensive documentation in the scientific hydroclimatic literature that (1) provides an overview of necessary requirements, statistical assumptions and consequential limitations of copulas, (2) clarifies some of the common misconceptions around copulas, and (3) offers clear guidelines on how to implement copulas, in particular for studies of future hydroclimatic impacts.

Accordingly, this paper aims at filling this gap and serves as an overview of the state of the art of using copulas in

hydroclimatology for practitioners interested in adopting this method for their research. We aim to illustrate a practical approach on copula-based modeling of hydroclimatic variables exemplified by precipitation and temperature. We introduce copula concept and bring forward a step-by-step copula approach (section 2). Based on a systematic literature review and a case study in Sweden we also explain common pitfalls and misconceptions (section 3), and provide a decision support framework for applying copulas (section 4) to support researchers and decision makers in addressing climatological hazards

and sustainable development.

## 2. Step by step copulas

### 2.1. What is a copula?

A copula is a mathematical function that couples marginal distributions and represents the joint probability distribution (Fig. 2). Copulas have been introduced in the early work of Sklar (1959), who showed that any multivariate joint cumulative

distribution function (CDF) could be specified as a copula function of the margins, assuming that the margins are continuous, time-independent and uniformly distributed on the [0, 1] interval. Therefore, if $H$ is an m-dimensional cumulative distribution function with one-dimensional margins denoted as $F_1, F_2, \dots, F_m$, then an m-dimensional copula (C) exists such that:

$$H(x_1, x_2, \dots, x_m) = C\big(F_1(x_1), F_2(x_2), \dots, F_m(x_m)\big) \tag{1}$$

for which $(x_1, x_2, \dots, x_m)$ are vectors of inputs.

If $F_1(x_1) = u_1$, $F_2(x_2) = u_2$ and $F_m(x_m) = u_m$, the copula density function, c can be derived as (Shojaeezadeh et al., 2018):

$$c(u_1, u_2, \dots, u_m) = \frac{\partial^d C(u_1, u_2, \dots, u_m)}{\partial u_1 \dots \partial u_m} \tag{2}$$

Then, the joint density function can be decomposed as:

$$f(x_1, x_2, \dots, x_m) = c(u_1, u_2, \dots, u_m) \prod_{k=1}^{m} f_k(x_k) \tag{3}$$

where $f_k$ denotes the density function of the $k^{\text{th}}$ dimension.





Although copulas can be defined for any number of variables $n$ (so-called n-dimensional copulas), two-dimensional copulas that link only two variables, such as temperature and precipitation, are much more common in hydroclimatic applications and therefore in the focus of this paper. In the following sections, we give a brief overview of both empirical and theoretical copula distributions.

### 2.2.  Empirical copula

To capture the underlying joint probability between two random variables, the joint rank correlation of the variables, termed *empirical copula* (Genest and Favre, 2007), needs to be  derived. The empirical bivariate copula is defined as the discrete function $C_n$ given by:

$$C_n\left(\frac{i}{n},\frac{j}{n}\right) = \frac{\{r_{1i}\leq R_1 \wedge r_{2j}\leq R_2\}}{n}$$  (4)

where $r_1$ and $r_2$ denote the order statistics of the sample and provides the cardinality of the subsequent set and $i$ and $j$ label the

individual sample elements. The result would be the joint CDF of uniformly distributed marginal (Charpentier et al., 2007). According to Eq. (4), the empirical copula can be interpreted as the cumulative distribution of rank-transformed data.

### 2.3.  Theoretical copula families

The governing pattern of the empirical copula can be used to guide decisions towards suitable theoretical copulas or so-called copula families. The two most widely used copula families in hydroclimatic modeling studies are *Elliptical* and *Archimedean*

copulas. These two families are different in complexity, tail behaviour, dependence structure and their theoretical parameter estimation method (Sadegh et al., 2017). In order to demonstrate how these families behave in shape and tail dependence, 1000 normally-distributed points with Pearson correlation coefficient, $\rho = 0.75$ were generated. Thereafter, the five most widely used copulas were fitted to the data (Fig. 3).

It should be noted that the parameters of bivariate copulas reflect the strength of mutual dependence between two variables

(e.g., between temperature and precipitation). To account for the degree of dependence between two variables, a critical assessment of their correlation is essential. The strength of dependence can be quantified through empirical measures, including Pearson's correlation coefficient ρ, Kendall's $\tau$, or Spearman's S.

*Pearson correlation* (Pearson 1920) represents the linear correlation between $x_1$ and $x_2$ as follows:

$$\rho_{x_1 x_2} = \frac{Cov(x_1,x_2)}{\sigma_{x_1}\sigma_{x_2}}$$  (5)

*Kendall's $\tau$* (Kendall 1938) and *Spearman's S* (Spearman 1904) are measures of rank correlation between the ranks of two variables, which can be expressed as:

$$\tau = \frac{number\ of\ concordant\ pairs - number\ of\ discordant\ pairs}{\frac{n(n-1)}{2}}$$  (6)

$$S = 1 - \frac{6\sum_1^n d_i^2}{n(n^2-1)}$$  (7)





where $d_i$ is the difference between two ranks and $n$ is the number of observations. These correlation measures can further be

used as a guide to estimate copula parameters for different copula families.

### 2.3.1. Archimedean copulas

Archimedean copulas are flexible tools to capture various dependence structures, e.g. concordance and tail dependence (Hofert, 2008). This makes them especially suitable for the modeling of extreme events.

The general form for Archimedean copulas is:

$$C(u_1, u_2; \theta) = \psi^{-1}(\psi(u_1) + \psi(u_2)), \ u \in [0, 1] \tag{8}$$

where $\theta$ is the selected Archimedean copula parameter, $\psi$ is defined on (0 1] is a continuous strictly decreasing convex function, known as the generator function and its pseudo-inverse $\psi^{-1}$ (Genest and Rivest, 1993).

Aloui, Ben Aïssa, and Nguyen (2013) distinguished between three different types of Archimedean copulas (Fig. 3):

*The Gumbel copula* (Gumbel 1960) is an asymmetric copula with higher probability concentrated in the upper tail. It is given

by:

$$C(u_1, u_2) = exp\left\{-\left[(-ln u_1)^\theta + (-ln u_2)^\theta\right]^{\frac{1}{\theta}}\right\} \tag{9}$$

*The Clayton copula* (Clayton 1978) has higher probability in the lower tail and is expressed as:

$$C(u_1, u_2) = \left(u_1^{-\theta} + u_2^{-\theta} - 1\right)^{\frac{-1}{\theta}} \tag{10}$$

*The Frank copula* (Frank 1979) tends to have lower density in the tails compared to Clayton and Gumbel copulas. Also, it is

capable of retaining negative correlations. Frank copula is defined as:

$$C(u_1, u_2) = \frac{-1}{\theta} ln\left(1 + \frac{(exp(-\theta u_1)-1)(exp(-\theta u_2)-1)}{exp(-\theta)-1}\right) \tag{11}$$

In the Archimedean copulas, the copula parameter $\theta$ can be derived from Kendall's $\tau$ (Allen et al., 2017). Depending on the type of Archimedean copula, parameter $\theta$ takes different values and has different relationships to Kendall's $\tau$ (Table 1).

After estimating $\theta$ based on Kendall's $\tau$ (Table 1), the copulas can be computed using their respective formulas.

### 180  2.3.2. Elliptical copulas

Elliptical copulas are widely used in hydroclimatic modeling, as they are symmetric and capable of representing positive and negative correlation. They are, however, less suitable to analyse and model hydroclimatic extremes, when there is evidence of a sharp tail dependence.

The governing function for Elliptical copulas is:

$$C(u_1, u_2) = \Phi\left(\Phi^{-1}(u_1), \Phi^{-1}(u_2)\right) \tag{12}$$

where $\Phi$ is a suitable multivariate distribution and $\Phi^{-1}$ denote the inverses of univariate marginal distributions. Several methods varying in accuracy and complexity have been suggested to estimate the associated copula parameters for Elliptical





copulas. For example, Aas (2004) suggested a Monte-Carlo-based method for parameter estimation, whereas Weiß (2011), used a Maximum likelihood approach.

The two most widely used types of Elliptical copulas are Gaussian and student-t copulas.

The *Gaussian copula* is expressed as:

$$C(u_1, u_2) = \Phi\big(\Phi^{-1}(u_1), \Phi^{-1}(u_2)\big) = \int_{-\infty}^{\Phi^{-1}(u_1)} \int_{-\infty}^{\Phi^{-1}(u_2)} \frac{1}{2\pi\sqrt{1-\theta^2}} \exp\left(-\frac{x_1^2 - 2\theta x_1 x_2 + x_2^2}{2(1-\theta^2)}\right) dx_1\, dx_2 \tag{13}$$

where $\Phi$ is the standard bivariate normal distribution with linear correlation coefficient $\theta$ and is restricted to the interval $(-1\ 1)$. There is no closed form to estimate the associated copula, but there are numerous ways to estimate it

numerically (Aloui et al., 2013).

The *Student-t copula is* given by:

$$C(u_1, u_2) = \Phi\big(\Phi^{-1}(u_1), \dots, \Phi^{-1}(u_2)\big)$$

$$= \int_{-\infty}^{t_v^{-1}(u_1)} \int_{-\infty}^{t_v^{-1}(u_2)} \frac{1}{2\pi\sqrt{1-\theta^2}} \left(-\frac{x_1^2 - 2\theta x_1 x_2 + x_2^2}{v(1-\theta^2)}\right)^{-\frac{v+2}{2}} dx_1\, dx_2 \tag{14}$$

The Student-t copula is characterized by a higher density in tails and an increased probability of joint extreme events compared

to the Gaussian copula (Aas, 2004). This is reflected by the added $v$ parameter, which represents the degrees of freedom. $t_v^{-1}(u)$ denotes the inverse of the CDF of the standard univariate Student-$t$ distribution. By increasing the value of $v$, the co-dependence strength in the tails will decrease.

Embrechts, Mcneil, and Strauman (1999) estimated the Elliptical copula parameter from the rank-based Kendall's $\tau$ or Spearman's $S$. Later, Aas (2004) showed that the co-dependence structure for Elliptical copulas can be presented by linear

(Pearson) correlation. Correspondingly, their copula parameter $\theta$ can either be estimated as being equal to the linear (Pearson) correlation or derived from Kendall' $\tau$ or Spearman's $S$. For more details on the corresponding equations, we refer the readers to Aas (2004).

### 2.4. Copulas as a basis for computing conditional distributions

In a bivariate case, copulas can be used to compute the conditional distribution of one variable given a specific value of second

variable. An example is the estimation of the conditional probability distribution of precipitation, given one particular temperature probability and the known joint probability distribution.

The conditional probability of $x_2$, given the occurrence of $x_1$, can be shown as:

$$f(x_2|x_1) = \frac{f(x_1, x_2)}{f(x_1)} \tag{15}$$

where $f(x_1, x_2)$ gives the joint density of $x_1$ and $x_2$, while $f(x_1)$ gives the marginal density of $x_1$. Considering Eq. (3) the

conditional copula function is reorganized as:

$$f(x_2|x_1) = \frac{c(u_1, u_2) f(x_1) f(x_2)}{f(x_1)} = c(u_1, u_2) f(x_2) \tag{16}$$





In practice, the conditional distribution can be obtained via Monte Carlo simulation, which is a numerical approach that relies on repeated random sampling (Aas, 2004). In order to compute the conditional probability from the copula, the following steps should be taken (Salvadori et al., 2007):

- Generate two independent sets of random variables $u_1$ and $r$ on (0,1) domain.
- Set $u_2 = c_{u_1}^{-1}(r)$
- Repeat until the latter is solved.

It is noteworthy to mention that in case of the Elliptical copulas, the conditional probability, can be calculated by Cholesky decomposition of the Elliptical copula parameter (Li and Babovic 2019).

**2.5. Performance measures for selecting a suitable copula family**

After fitting copulas to the data, some performance measures are used to analyze the goodness of fit obtained from different copula families and parameters. This can be done by comparing cumulative distributions of empirical and theoretical copulas either parametrically or non-parametrically (Genest et al., 2009). The Cramér von Mises ($S_n$) (Eq. 17) and Kolmogorov Smirnov ($D_n$) (Eq. 18) statistics are two of the widely used performance measures for copula parameter estimation (Madadgar

and Moradkhani, 2014).

$$S_n = \sum_{i=1}^{n} |C_n(x) - C_\theta(x)|^2 \qquad (17)$$

where $C_n$ is the Empirical joint distribution and $C_\theta$ is the fitted copula.

$$D_n = \sup|C_n(x) - C_\theta(x)| \qquad (18)$$

in which $sup$ is the supremum of the distances at point $x$.

However, it is also important to evaluate the p-values of the applied performance measures (Berg, 2009; Genest et al., 2009). The p-value of Cramér von Mises or Kolmogorov Smirnov can be found by bootstrapping via Monte Carlo approach by following the steps proposed by Genest, Rémillard, and Beaudoin (2009). The null hypothesis of the test is the acceptance of the parametric copula. For a particular copula, the p-value is sufficient to determine the acceptance or rejection of the null hypothesis with the significance level of $\alpha$, but in a group of different acceptable copulas, the best alternative is the one with

the smallest $S_n$ or $D_n$ and the greatest p-value (Madadgar and Moradkhani, 2014).

In assessing the performance of copula models, using only one performance measure can be misleading, therefore it is recommended to use an ensemble of measures (Sadegh et al., 2017). The Nash-Sutcliffe Efficiency ($E_{NS}$) (Eq. 19) and Root Mean Square Error ($E_{RMS}$) (Eq.20) are two widely used performance measures in hydroclimatic applications, which can be adopted in copula model selection as well.

$$E_{NS} = 1 - \frac{\sum_{i=1}^{n} |C_n(x) - C_\theta(x)|^2}{\sum_{i=1}^{n} |C_n(x) - \bar{C}_n(x)|^2} \qquad (19)$$

$$E_{RMS} = \sqrt{\frac{\sum_{i=1}^{n} |C_n(x) - C_\theta(x)|^2}{n}} \qquad (20)$$





## 3. Common issues, misconceptions and pitfalls

To identify common issues, misconceptions and pitfalls in the application of copulas in the existing hydroclimatic literature, a systematic literature review was conducted. We reviewed all articles returned from the Scopus search engine using the
keywords "Copula + Precipitation + Temperature". As of 1ˢᵗ of June 2020, these keywords resulted in a list of 85 papers, of which 16 were not accessible due to missing permissions. To examine the applied copula methods and underlying assumptions, the remaining 69 publications were investigated to evaluate how physical characteristics had been considered in these publications and if they fulfilled the statistical requirements necessary to adopt a copula framework. In order to do so, the following six aspects were assessed in detail.

*Physical characteristics:*

    1- What is the **spatiotemporal scale** of the study?

    2- Does the paper analyse and interpret the **correlation** between temperature and precipitation?

*Statistical requirements:*

    3- Does the paper consider **autocorrelation** within each of the variables (i.e., autocorrelation of temperature respective
260       to autocorrelation of precipitation)?

    4- Does the paper consider zero-precipitation, e.g. by introduction of a **precipitation threshold** and deals with **data with the same rank**?

    5- Does the paper reflect on the **significance of correlation** and **stationarity** issue?

    6- Are several copula families fitted and **checked for errors**?


For each of these aspects, we first explain its significance. Thereafter, we demonstrate potential consequences, if not properly addressed, based on a real hydroclimatic case, namely the Vattholma river catchment in Southcentral Sweden (Fig. 4) close to the city of Uppsala and approximately 70 km north of the capital Stockholm. For this catchment, we constructed a copula of observed temperature and precipitation data for the climate normal periods 1961-1990. Daily precipitation and temperature
data were downloaded from a freely available database ([www.vattenwebb.smhi.se](www.vattenwebb.smhi.se)) provided by the Swedish Meteorological and Hydrological Institute (SMHI). According to this data, the catchment is characterized by a warm-summer continental climate with an average annual temperature of 5.2°C and an annual precipitation sum of 633 mm.

### 3.1. Spatiotemporal scale

First, each paper was scanned for the spatial scale of the investigated study site(s). Roughly 56% of the reviewed papers were
based on meso- to large-scale studies (i.e., $> 300\ km^2$), while the remaining 44% were conducted on small scales (i.e., $< 300\ km^2$). The spatial scale is of importance, because it affects some of the statistical requirements of copulas that are explained further below.





Our showcase catchment Vattholma has an area of 293 $km^2$, thus, can be considered small-scale. The catchment is predominantly flat with altitudes ranging from 25 to 65 meters above sea level. It is mostly covered by forest (81%), while the
remaining area includes 10% lakes/wetlands, 7% open land and 2% populated areas (Fig. 4).

In the second step, we checked for the temporal resolution (i.e., hourly, daily and monthly) of the published literature. We found that only 3% (2 publications) considered hourly values of temperature and precipitation. Most (66%) used daily values, whereas 31% considered monthly values. The temporal resolution, similar to the spatial scale, is crucial when it comes to some of the statistical properties such as correlation and autocorrelation.

**3.2. Correlation**

Copulas are built on the co-dependence between variables and it is crucial to examine the correlation between the variables (Eqs. 5-7). For precipitation and temperature, the co-dependence often varies regionally and depends on the spatial scale at which the data are recorded. It is also important to check for significance of correlation because it allows to examine if correlation strength is independent from the selected sample. But it is not always possible to find a significant statistical
correlation between precipitation and temperature at a given scale (Serinaldi, 2016).

Out of the 69 reviewed papers, 29% did not indicate the strength or significance of their correlation by any of the aforementioned correlation measures. 25% mentioned that their data shows some degree of correlation, but did neither specify the strength nor the significance nor any copula parameters calculated based on the correlation value. Less than half of all papers (46%) transparently stated the correlation strength and represented the correlation of the data either graphically or with
one of the dependence empirical measures. Only 23% presented the strength and discussed the statistical significance of the correlation, e.g. by checking the p-value of their correlation measure.

For the Vattholma catchment, the correlation strength depends on the temporal resolution. We also checked the correlation significance and demonstrate that in case of Vattholma catchment there are months in which the correlation is not significant at the 5% level, thus cannot be considered as reliable correlation between variables. Furthermore, in the daily resolution the
correlation is generally weaker than monthly resolution (Fig. 5(a), Lines 1 and 2). As the Pearson correlation was weaker during winter, we considered the co-dependence structure to be addressable in the summer. We choose to use the daily resolution to further generate copula co-dependence structure, because daily data can readily be used for finer resolution hydrological purposes (e.g., bias correction, daily runoff estimation).

**3.3. Stationarity of correlation**

Copulas can also be used for multivariate frequency analyses to calculate joint return periods of extreme variables as e.g. rainfall intensity and duration of individual precipitation events or other variables (e.g., Salvadori and De Michele 2010 or AghaKouchak et al. 2014). In these cases, copulas are established for a set of available data within a certain time frame to draw conclusions about the frequency of certain events that depend on the combined influence of two variables such as temperature and precipitation in case of droughts. However, in these cases stationarity is of importance, because findings from





such an analysis are commonly used for future risk analyses. Whether or not this underlying stationarity assumption is actually met, is subject to debate (e.g., McCarl, Villavicencio, and Wu 2008; Vaze et al. 2010 or Lins and Cohn 2011). As McCarl et al. (2008) framed it, stationarity implies that "natural systems fluctuate within an unchanging envelope of variability". However, given the scale of ongoing climate change and the projected changes in the climate system that we are likely to face in the future, the assumption of stationarity is possibly not met, which could affect the applied risk measures (Cooley, 2013).

In the case of copulas, if the underlying idea is to draw conclusion for potential futures, it is crucial to check for changes in the correlation strength over time and the statistical significance. In other words, when the correlation is highly sensitive to the selected time period, it is an indication of a non-stationary behaviour. Nevertheless, we found only 5 of all 69 reviewed papers (7%) touching upon the assumption of stationarity of correlation or discussing changes in correlation over time.

To check if the correlation was stationary in our Vattholma showcase, we followed a random selection technique: We first
selected a sample of 10 random years (preserving the order of days within a year but without necessarily selecting consecutive years) from the 30 years period. Thereafter, we computed the correlation between daily resolution variables within each month in the chosen sample. We repeated this procedure 1000 times and found that in many months, correlation was changing significantly across different sets of 10 years (Fig. 5(b)). For example, in October and November, the correlation varied from positive to negative values. We also assessed the significance of the correlation in each month by checking the average p-value
of the 1000 samples. In June and July, the correlation had the lowest average p-value and is relatively strong, therefore it could be concluded that correlation within these months was significant. We adopted a copula framework on July, because it has significant correlation at both daily and monthly resolutions (Fig. 5(a)-(b)).

### 3.4. Autocorrelation

Autocorrelation describes how time series are dependent on a delayed copy of themselves. According to Cong and Brady
(2012), the presence of autocorrelation reduces the efficiency of copula, because it boosts the variances of residuals and estimated coefficients. In order to benefit from the copula concept, variable sets should be *time-independent* (i.e., not feature autocorrelation) (Mao et al., 2015).

In some studies, copulas have been applied to autocorrelated data using so-called serial copulas (Genest and Rémillard, 2004; Ghoudi and Rémillard, 2004) or autocopulas (Rakonczai et al., 2012). But the mathematical procedure to drive these copulas
is quite demanding and further uncertainties may emerge (Rakonczai et al., 2012). In contrast, other papers argued that copulas are still applicable in case of only low-degree (removable) auto-correlation in the time series (Laux et al., 2011; Vogl et al., 2012).

Therefore, it is important to consider the degree of autocorrelation in the studied data. However, 75% of the reviewed papers did not mention if their data exhibited autocorrelation. The remaining 25% did check for autocorrelation, but only about half
of those addressed it by removing it or by bootstrapping.

For the Vattholma catchment, the autocorrelation function (ACF) for precipitation (Fig. 6(a)) and temperature (Fig. 6(b)) in July indicates some autocorrelated behavior. In order to account for autocorrelation, Laux et al. (2011) suggested using a first-





order autoregressive model AR(1) to synthesize time-dependence in the time series. Then the synthesized autocorrelated data is subtracted from the original set. We followed this procedure to create a data set free from autocorrelation, which we validate
by plotting the ACF of the residuals (Fig. 6(c)-(d)), indicating that the data is now time-independent and can be used in the copula approach.

### 3.5.  Correlation for data with the same rank (dry days)

Some studies have concentrated on the joint distribution of precipitation and temperature on wet days (i.e., Piani and Haerter 2012; Li et al. 2014; Räty et al. 2018). In other words, the joint distribution has been constructed between precipitation values
larger than zero and the associated temperature on the same day. As mentioned earlier, copulas must be built on continuous sets of variables, which is shown in Eq. (4) and is also reflected in the method for constructing empirical copula, which is based upon rank dependence between variables (see section 2.2). Therefore, data points with the exact same precipitation (i.e., most commonly dry days with zero precipitation) are problematic as they lead to several ranks associated with temperature but only one rank for zero precipitation. Depending on the region studied, there can be a considerable number of zero-
precipitation (dry) or drizzle days. Therefore, a precipitation threshold needs to be defined, below which all data points should be removed. Removing such data can often eliminate a large portion of the precipitation-temperature data set, especially in drier regions. Accordingly, some efforts have been made to control data which have the same rank by adopting some statistical methods. For instance, Salvadori, Tomasicchio, and D'Alessandro (2014) suggested a jittering algorithm before computing the copula parameters, in order to perturb the data that have the same rank. Regardless of the chosen method to deal with data
with the same rank, an added post-processing step is probably needed that also handles the temperature data points that are not considered in the construction of the copulas because they fall below the precipitation threshold. This can be considered as a source of added uncertainty, especially if the data is to be later used for bias correction or crop yield estimation. Therefore, we suggest that each published study transparently explains what part of the data has been used for constructing the copula (e.g., only days with precipitation above zero or some threshold) and how the remaining data has been treated.
However, our literature review revealed that 76% of the publications did not mention this particular issue. Only 24% of the studies transparently stated that dry/drizzle days are excluded, and only 9% explained how the remaining dry days were handled. For example Li and Babovic (2019) later handled the eliminated data points by scaling the associated temperature distribution to the full range of temperature data. This can be considered an applicable alternative, but the added uncertainties should not be neglected. Of course, it should be mentioned that 21% of the reviewed papers focused on the extreme part of the
joint distribution, therefore there was no need to focus on the complete range of data and the effect of neglected variables.

### 3.5.1. Influence of geography, spatial scale and temporal resolution

It should be noted that whether or not the data includes zero-precipitation or drizzle days is largely dependent on the study region, the spatial scale and the temporal resolution, e.g. arid regions or smaller domains with few measurement stations. In arid regions the occurrence of data points with no precipitation is considerably higher. Similarly, at higher temporal resolution





(e.g., hourly or daily), more data points will fall below the threshold. For example, in the Vattholma catchment, the number of wet days varied from month to month, ranging from 42% to 64%. We here applied a precipitation threshold of 0.1 mm to remove drizzle days.

Excluding zero-precipitation or drizzle days affects the correlation between precipitation and temperature. Since a significant correlation is the basis for applying copulas, it is essential to re-check the correlation after removal. In our showcase Vattholma,

for daily resolution data, correlation in the entire 30-year period changed only for some months after removing drizzle days (Fig. 5(a), line 3). The correlation in June-August was highly affected by removing days without precipitation as there were more dry days in these months.

### 3.6.  Selecting suitable copula families

As mentioned earlier, in order to assess the copula performance, a validation of the copula model compared to the empirical

joint distribution is vital. In the reviewed literature, 15% of the papers used the empirical copula as the co-dependence structure, therefore it was not necessary to validate the fitted copula families. However, 28% of all papers fitted a copula on the data without documenting any performance measures. 57% checked for the goodness of fit of different copulas using at least one performance measure.

We here demonstrate how such performance measures can be applied practically. For the Vattholma case study, different

copula families were adopted to construct the joint copula between daily temperature and daily precipitation data in July. Since these variables were negatively correlated, the Student-t, Gaussian and Clayton copulas were selected as potentially suitable copula families, because they are capable of addressing negative correlation as well. Thereafter, the fitted copulas were visually compared to the empirical copula at (0 1] scale (Fig. 7(a)).

After transforming to the original scale, corresponding precipitation and temperature and their density plots were estimated

(Fig. 7(b)). Thereafter, we computed the earlier-described set of performance measures: $E_{RMS}$, $E_{NS}$, $S_n$ and $D_n$ statistic (Table 2). We also checked for the significance of calculated $S_n$ at a significance level of $\alpha = 0.05$. Three of these measures (i.e., $E_{RMS}$, $E_{NS}$ and $D_n$) resulted in the same ranking of the copulas, namely the Student-t copula having the best fit and the Clayton having the worst. Only the $S_n$ criterion ranked the Gaussian copula first (Table 2).

Here, evaluation of copulas by $E_{RMS}$ or $E_{NS}$ did not reveal the weak performance of Clayton copula that is shown by the other

performance measures, underlining the need of assessing the goodness of fit by multiple performance measures.

### 4.  Decision support framework for applying copulas

Properly accounting for the joint distribution of precipitation and temperature is of great importance for a range of different hydroclimatic applications. Yet, we highlighted a number of common issues, misconceptions and pitfalls that need careful handling to ensure an adequate, consistent and transparent use of copulas in hydroclimatology. Below, we briefly outline a

stepwise decision support framework (Fig. 8) for hydroclimatic copula applications with the goal of preventing users from stepping into the many statistical traps that copulas conceal. The proposed decision support framework has been developed to



minimize the incorrect application of copulas and to ensure reliable and comparable results, by guiding the end user through six generic steps (Fig. 8) that should typically be adopted before and during implementation of the copula approach, and by elucidating different alternatives that can be undertaken.

For each step in the framework (Fig. 8), in-depth explanations, mathematical equations, common pitfalls and scientific references for the various associated tasks can be found in the previous sections. A short narrative of each step is provided here: Prior to constructing bivariate copulas, it is worth reflecting on whether the two involved variables are controlled by related driving forces that could, at a relevant spatiotemporal scale, lead to a dependence structure between them (section 3.1). If so, their correlation can further be assessed with a particular focus on underlying key assumptions (section 3.2). In cases

where stationarity is relevant, it is crucial to investigate the correlation over time (section 3.3). Prior to implementation of the copula approach, it is further vital to test for autocorrelation of the variables and potentially remove any existing serial correlation (section 3.4). To construct an empirical copula based upon rank dependence between variables, the variables need to be sorted and ranked, which allows to detect data points with the same ranks that cannot be included when constructing the copula and that need to be handled separately (section 3.5). Thereafter, the correlation value can be used to estimate copula

parameters for different families. The fitted copula families are then compared to the empirical copula with help of one or more performance measure to decide on the best-fitting copula family and type (section 3.6).

## 5. Concluding remarks

This paper provides an overview of the state of the art of using copulas in hydroclimatology and highlights necessary requirements, statistical assumptions and limitations of using copulas in hydroclimatic studies. Based on a systematic literature

review, we identified common pitfalls and misconceptions, and highlighted ways to avoid them by providing a showcase study. We would like to mention that copula modeling, similar to any other modeling effort, is associated with a range of uncertainties stemming from observation errors, limited observations, and copula model structural and parameter uncertainty, among others, that translate into joint probability estimates and other derived inferences that should be considered (Sadegh et al., 2018).

In our study, two general issues emerged. Firstly, important steps for deriving copulas (e.g., handling of precipitation-free

days) are often not reported, which makes many studies not reproducible. Secondly, key statistical assumptions required by copulas (e.g., significance and stationarity of correlation or absence of autocorrelation) are often not tested, which could severely undermine the credibility of certain study conclusions. Considering the growing interest in copulas in hydroclimatic applications, we emphasize the need to address these issues and provide a visual decision support framework to foster the transparency and reproducibility of scientific publications, while at the same time reducing the risk for inconsistent or even

incorrect use of copulas in hydroclimatic research.

Although the framework has been specifically developed for temperature-precipitation copulas, it is anticipated that it can be easily applied to a wide range of hydroclimatic research studies. By breaking down the process into six generic steps and providing short and simple narratives, the framework can to a large extent easily be adopted for other pairs of (hydroclimatic)



variables to construct bivariate copulas, thereby supporting researchers and decision makers in addressing climatological

hazards and sustainable development.

*Code and data availability.* The codes are developed in MATLAB and are publicly available at https://doi.org/10.5281/zenodo.3900001. The observational data for daily precipitation and temperature in the climate normal periods 1961-1990 were downloaded from a freely available database (www.vattenwebb.smhi.se).

*Author contribution.* FT and CT designed the framework. FT developed the code. FT, CT and TG designed the visualization of the article. JOH and MS contributed to the statistical requirements of the framework. JOH, TB and OR contributed to the hydroclimatic implications of the results. FT developed the first manuscript. All co-authors contributed to interpret the final results. All co-authors contributed to edit the manuscript.

*Competing interests.* The authors declare that they have no conflict of interests.

*Acknowledgements.* This research was supported by the Swedish Research Council (VR, grant no. 2017-04970).



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





**Figures**

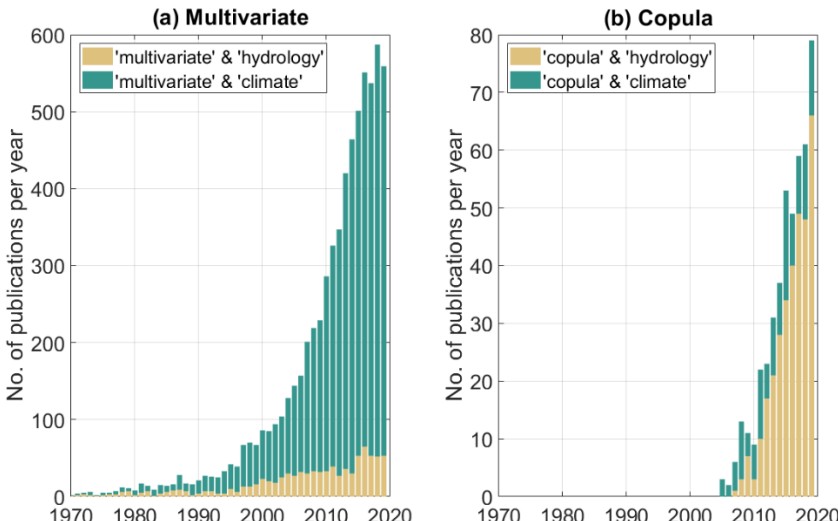

**Figure 1. Number of annually published peer-reviewed papers listed by Scopus (a) when combining the search term 'multivariate' with either 'hydrology' (beige bars) or 'climate' (turquoise bars), and (b) when combining the search term 'copula' with either**
**'hydrology' (beige bars) or 'climate' (turquoise bars).**

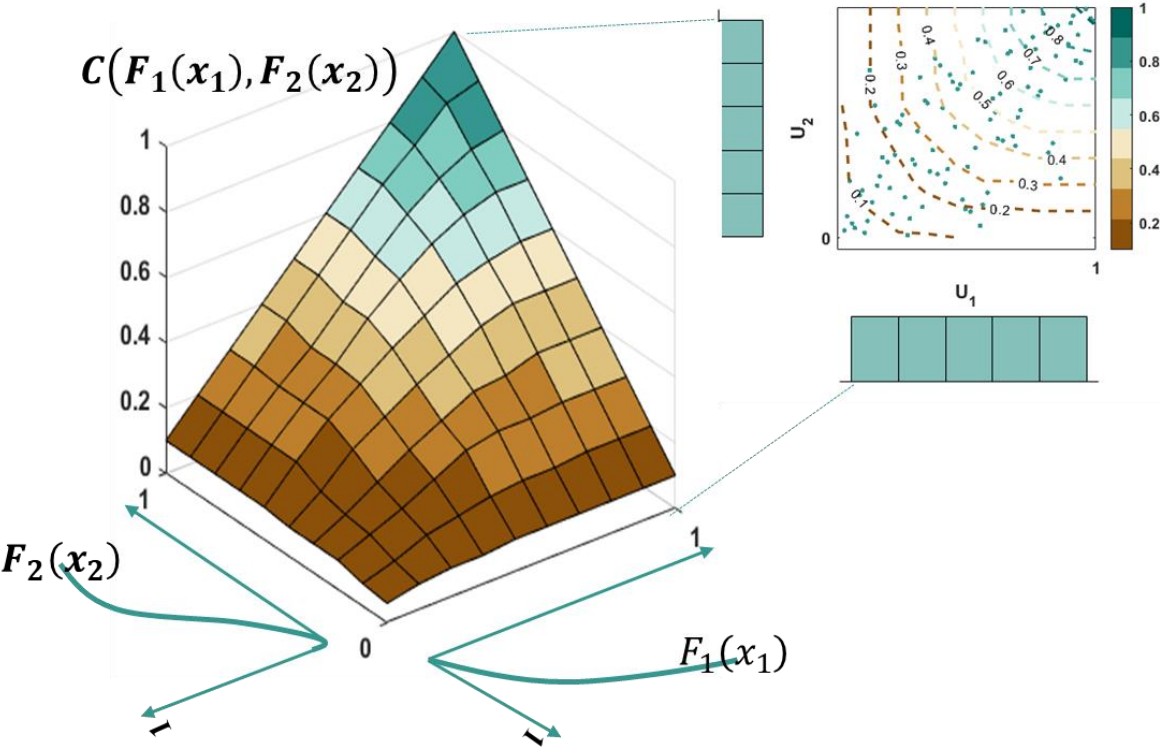

**Figure 2. Schematic visualization of the empirical two-dimensional copula (surface) and its relation to the marginal cumulative distribution functions.**



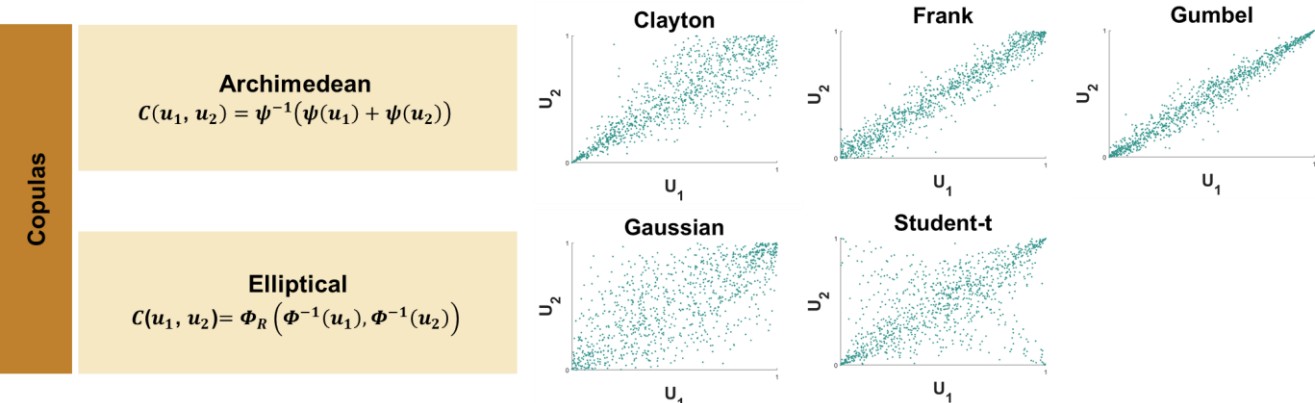

**Figure 3. Illustration of two widely used copula families (Archimedean and Elliptical) and their common sub-types. 1000 normally-distributed points with Pearson correlation of $\rho$=0.75 were randomly generated and transformed into uniform margins $U_1$ and $U_2$ and plotted in two-dimensional space.**



**Figure 4. Overview of the Vattholma catchment, with its outlet being located at 60.0194°N and 17.7307°E.**



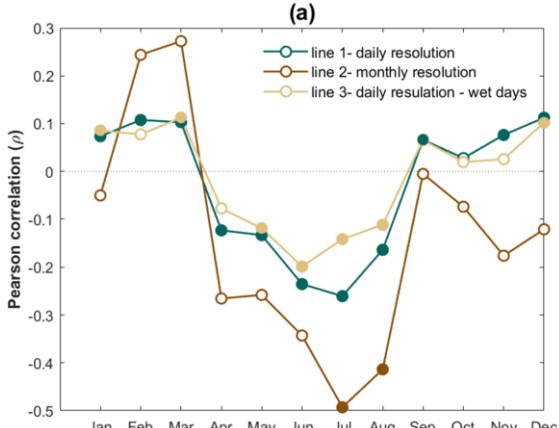
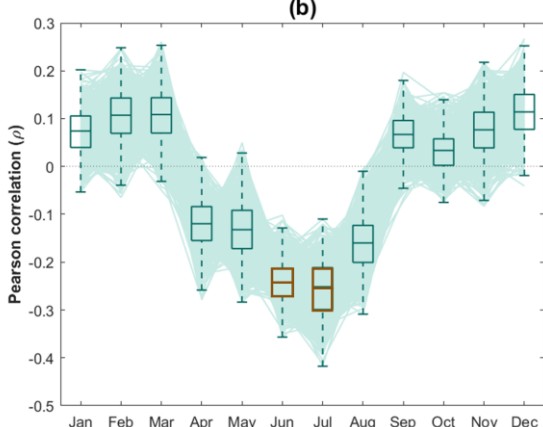

**Figure 5. Linear correlation between temperature and precipitation: (a) Comparison of different temporal resolutions. Filled markers indicate significance at 0.05% significance level. (b) Daily resolution correlation based on 1000 samples of 10 randomly drawn years from a 30 years period. Boxes in brown represent months with the lowest average p-value (here below 0.001).**





**Figure 6. The autocorrelation function (ACF) for daily precipitation (a) and temperature (b) in July compared to the precipitation residuals (c) and temperature residuals (d) after removing autocorrelation from the data.**





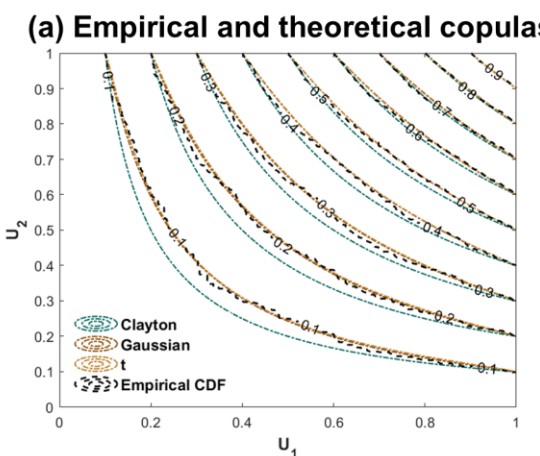

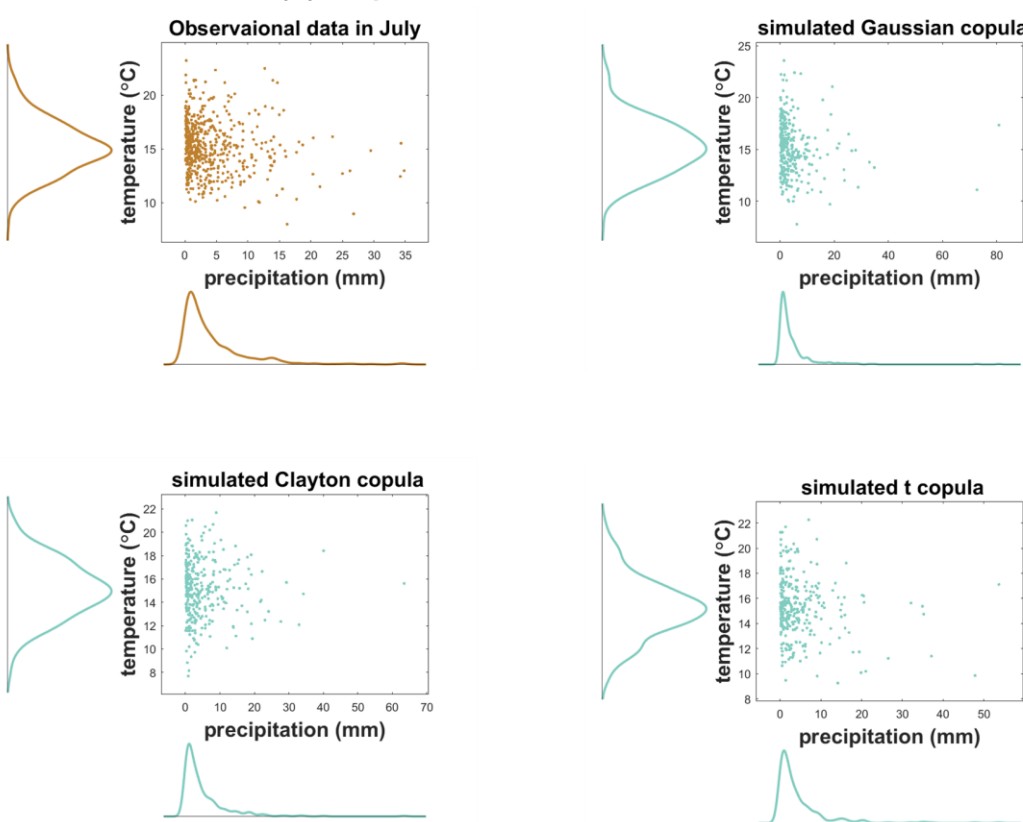

**Figure 7. Copulas for daily precipitation and temperature in July in the Vattholma catchment. (a) Comparison of three fitted copulas with the joint empirical cumulative distribution function (CDF) of transformed values. The numeric labels on the curves correspond to the cumulative joint probability of $U_1$ and $U_2$. (b) Joint copula distribution of the original data and different copula families. Each panel shows a scatterplot of daily precipitation (mm) and daily average near-surface temperature (°C) and the associated marginal probability density function.**





**Figure 8. Decision support framework for adopting the copula approach. Each step (beige boxes) refers to a subheading of section 3, which brings up the statistical requirements and general assumptions that should be fulfilled.**





670

**Tables**

**Table 1. Archimedean copula parameter range and relationship to Kendal's τ.**

|  | θ | τ |
|---|---|---|
| Gumbel | $[1, +\infty)$ | $= \dfrac{\theta - 1}{\theta}$ |
| Clayton | $[0, +\infty)$ | $= \dfrac{\theta}{\theta + 2}$ |
| Frank | $(-\infty, +\infty) \backslash 0$ | $= 1 - \dfrac{4}{\theta}\left[\dfrac{1}{\theta}\int_0^\theta \dfrac{t}{e^t - 1}\,dt - 1\right]$ |





**Table 2. Different copula families and their performance measures as well as their ranking (numbers in parenthesis) in being able to represent the empirical copula. Families with the highest rank are shown in bold.**

| Performance measure | copula family | | |
| --- | --- | --- | --- |
| | Gaussian | Student-t | Clayton |
| $E_{RMS}$ (ranking) | 0.3252 (2) | **0.318** (1) | 0.4062 (3) |
| $E_{NS}$ (ranking) | 0.9956 (2) | **0.996** (1) | 0.9931 (3) |
| $S_n$ (ranking) | **8.2100 (1)** | 9.5648 (2) | 62.171 (3) |
| $D_n$ (ranking) | 0.0199 (2) | **0.0197** (1) | 0.0341 (3) |