# Peer review of "Copulas for hydroclimatic applications - A practical note on common misconceptions and pitfalls"

_Hydrology and Earth System Sciences, 2020_

## Short Comment (SC1) · 27 Jul 2020

Introduction

This is a particularly interesting work that concerns a very active topic of research in the hydrological domain (and beyond). Below there are a few comments that I hope the Authors might find useful, aiming to improve the quality of the manuscript, as well as better highlight some common misconceptions and pitfalls that regard particularly the case of Gaussian copula.

Comments

1. L88-89. The Authors write: "Since the early 2000's, copula methods have been adopted in hydrological modeling, which was triggered by the study of Salvadori and De Michele (2010)." With above sentence in mind I would like to bring to the Authors attention the works of Favre et al. (2004) and Salvadori and De Michele (2004), which if I am not mistaken are the first applications of copulas in hydrological domain (chronologically preceding the one already mentioned in the manuscript).

2. L90-101. In this paragraph the Authors mention numerous works that have used the notion of copulas for the development of various methods in hydrological domain. In this extent I think that it is useful to mention that copulas have also been used for the generation of synthetic hydroclimatic data, such as synthetic time series of rainfall, runoff, etc. (an important task required by many uncertainty-aware methods/models driven by stochastically-generated data). As in the case of random variables and multivariate distributions, also in this case copulas offer the necessary flexibility for modelling/simulation of non-Gaussian processes. For instance see the works of Lee and Salas (2011), Chen et al. (2015) and Hao and Singh (2013), as well as recent approaches in hydrological domain, based on the Gaussian copula (a construct related with the Nataf's joint distribution; see Lebrun and Dutfoy (2009), and references below for a discussion in a hydrological context) that allow the parsimonious simulation of multivariate stationary and cyclostationary processes with any marginal distribution and correlation structure (Kossieris et al., 2019; Tsoukalas et al., 2020, 2018a, 2018b) - also in a multi-scale/disaggregation context (Tsoukalas et al., 2019).

3. L204-205. With reference to Elliptical copula (i.e., the Gaussian and Student-t copula), the Authors write: "Later, Aas (2004) showed that the co-dependence structure for Elliptical copulas can be presented by linear (Pearson) correlation. Correspondingly, their copula parameter $\theta$ can either be estimated as being equal to the linear (Pearson) correlation or derived from Kendall' $\tau$ or Spearman's ðĺŚȨ. For more details on the corresponding equations, we refer the readers to Aas (2004)." Indeed there are relationships that link the Pearson's correlation coefficient with Kendall's and Spearman's

rank-based correlation coefficients, yet as highlighted in Tsoukalas et al. (2018b), section 3.2.3, these are valid if and only if both the marginals, and the copula are Gaussian (see also references therein).

When the copula is Gaussian, and the marginals are not (which is typical in hydrology), these relationships are no longer valid. In fact, in such cases, the Pearson's correlation coefficient depends on the marginals; since it involves the first cross-product moment among the variables (i.e., it involves the term E[X1, X2]), while the Kendall's and Spearman's correlations do not (since they are rank-based measures of dependence). In the case of Gaussian copula and non-Gaussian marginal, there is a non-analytical relationship that links the Pearson's correlation coefficient in Gaussian (in the manuscript's notation, the Gaussian copula parameter $\theta$) and target domain that has to be found by resolving of a double infinite integral. In particular, and with reference to hydrological domain, see Tsoukalas et al. (2020, 2019, 2018a, 2018b) and references therein.

In my view, the above are delicate, often neglected, points that concern the Gaussian copula, and therefore should be made clear in the manuscript, since they are both (very) common misconceptions/pitfalls that concerns the later (widely-used) copula.

4. L310-318. In this paragraph, as well as in other parts of the manuscript, the Authors discuss the debate between stationarity and non-stationarity. On this topic, and beyond the work of Lins and Cohn (2011), already cited in the manuscript, my suggestion to the Authors would be to review, (and cite if it is considered appropriate), the recent works of, Serinaldi et al. (2018), with emphasis on section 4.2, Koutsoyiannis and Montanari, (2007), (2015), Lins and Cohn (2011), Matalas (2012), and Montanari and Koutsoyiannis (2014). All these works discuss the importance of the assumption of stationarity, highlighting that it is an essential tool for inferencing from data (e.g., model fitting). See also the very interesting, note of Harry F. Lins[1], which concludes as follows:

Stationarity $\neq$ static

Non-stationarity ≠ change (or trend)

In my view, stationarity should not be viewed as a shortcoming, nor considered dead. It is recalled that non-stationarity implies non-ergodicity, which in turn makes inference from observed data impossible, unless of course the deterministic dynamics of the process (and hence potential change) are known; which in my understanding, is never the case in hydrological sciences.

Regards,

Ioannis Tsoukalas

PS. For convenience, please see the attached PDF file.

[1] http://www.wmo.int/pages/prog/hwrp/chy/chy14/documents/ms/Stationarity_and_Nonstationarity.pdf

References

Chen, L., Singh, V.P., Guo, S., Zhou, J., Zhang, J., 2015. Copula-based method for multisite monthly and daily streamflow simulation. J. Hydrol. 528, 369–384. https://doi.org/10.1016/j.jhydrol.2015.05.018

Favre, A., El Adlouni, S., Perreault, L., Thiémonge, N., Bobée, B., 2004. Multivariate hydrological frequency analysis using copulas. Water Resour. Res. 40. https://doi.org/10.1029/2003WR002456

Hao, Z., Singh, V.P., 2013. Modeling multisite streamflow dependence with maximum entropy copula. Water Resour. Res. 49, 7139–7143. https://doi.org/10.1002/wrcr.20523

Kossieris, P., Tsoukalas, I., Makropoulos, C., Savic, D., 2019. Simulating Marginal and Dependence Behaviour of Water Demand Processes at Any Fine Time Scale. Water 11, 885. https://doi.org/10.3390/w11050885

Koutsoyiannis, D., Montanari, A., 2015. Negligent killing of scientific concepts: the stationarity case. Hydrol. Sci. J. 60, 1174–1183. https://doi.org/10.1080/02626667.2014.959959

Koutsoyiannis, D., Montanari, A., 2007. Statistical analysis of hydroclimatic time series: Uncertainty and insights. Water Resour. Res. 43, 1–9. https://doi.org/10.1029/2006WR005592

Lebrun, R., Dutfoy, A., 2009. An innovating analysis of the Nataf transformation from the copula viewpoint. Probabilistic Eng. Mech. 24, 312–320. https://doi.org/10.1016/j.probengmech.2008.08.001

Lee, T., Salas, J.D., 2011. Copula-based stochastic simulation of hydrological data applied to Nile River flows. Hydrol. Res. 42, 318–330. https://doi.org/10.2166/nh.2011.085

Lins, H.F., Cohn, T.A., 2011. Stationarity: Wanted dead or alive? J. Am. Water Resour. Assoc. https://doi.org/10.1111/j.1752-1688.2011.00542.x

Matalas, N.C., 2012. Comment on the Announced Death of Stationarity. J. Water Resour. Plan. Manag. 138, 311–312. https://doi.org/10.1061/(ASCE)WR.1943-5452.0000215

Montanari, A., Koutsoyiannis, D., 2014. Modeling and mitigating natural hazards: Stationarity is immortal! Water Resour. Res. 50, 9748–9756. https://doi.org/10.1002/2014WR016092

Salvadori, G., De Michele, C., 2004. Frequency analysis via copulas: Theoretical aspects and applications to hydrological events. Water Resour. Res. 40. https://doi.org/10.1029/2004WR003133

Serinaldi, F., Kilsby, C.G., Lombardo, F., 2018. Untenable nonstationarity: An assessment of the fitness for purpose of trend tests in hydrology. Adv. Water Resour. https://doi.org/10.1016/j.advwatres.2017.10.015

Tsoukalas, I., Efstratiadis, A., Makropoulos, C., 2019. Building a puzzle to solve a riddle: A multi-scale disaggregation approach for multivariate stochastic processes with any marginal distribution and correlation structure. J. Hydrol. 575, 354–380. https://doi.org/10.1016/j.jhydrol.2019.05.017

Tsoukalas, I., Efstratiadis, A., Makropoulos, C., 2018a. Stochastic Periodic Autoregressive to Anything (SPARTA): Modeling and Simulation of Cyclostationary Processes With Arbitrary Marginal Distributions. Water Resour. Res. 54, 161–185. https://doi.org/10.1002/2017WR021394

Tsoukalas, I., Kossieris, P., Makropoulos, C., 2020. Simulation of Non-Gaussian Correlated Random Variables, Stochastic Processes and Random Fields: Introducing the anySim R-Package for Environmental Applications and Beyond. Water 12, 1645. https://doi.org/10.3390/w12061645

Tsoukalas, I., Makropoulos, C., Koutsoyiannis, D., 2018b. Simulation of Stochastic Processes Exhibiting Any‐Range Dependence and Arbitrary Marginal Distributions. Water Resour. Res. 54, 9484–9513. https://doi.org/10.1029/2017WR022462

Tsoukalas, I., Papalexiou, S., Efstratiadis, A., Makropoulos, C., 2018c. A Cautionary Note on the Reproduction of Dependencies through Linear Stochastic Models with Non-Gaussian White Noise. Water 10, 771. https://doi.org/10.3390/w10060771

Please also note the supplement to this comment: https://hess.copernicus.org/preprints/hess-2020-306/hess-2020-306-SC1-supplement.pdf

---

## Referee Comment (RC1) · Anonymous Referee #1 · 17 Aug 2020

**Review for manuscript "Copulas for hydroclimatic applications – A practical note on common misconceptions and pitfalls"**

**Authors:** Faranak Tootoonchi, Jan Olaf Haerter, Olle Räti, Thomas Grabs, Mjtaba Sadegh, and Claudia Teutschbein
**Journal:** Hydrology and Earth System Sciences

**Summary**

The study by Tootoonchi et al. lists and discusses pitfalls related to the multivariate analysis of precipitation and temperature using copulas. To identify pitfalls, they look at a sample of published manuscripts analyzing precipitation (P) and temperature (T) within a copula framework. To illustrate them, they fit different copulas to daily July precipitation and temperature for a case study in Sweden. They finally present a 'decision support framework' for the application of copulas to precipitation and temperature datasets consisting of the following steps: (1) determination of spatio-temporal scale, (2) computation of correlation between variables, (3) testing for stationarity, (4) testing for autocorrelation, (5) testing for ties, and (6) copula fitting and evaluation.

**General comments**

I agree with the authors that copulas are often used in the fields of hydrology and climatology and that there is a lot of room for improvement in when and how they are applied. In my point of few, the most common 'pitfalls' are that (1) the nature of dependence is often not studied before starting to test various copulas; (2) the dependence structure is often reduced to correlation, (3) no proper goodness-of-fit tests are applied to reject inappropriate copulas. While the authors detect several other pitfalls related to P-T analyses, these important pitfalls are not addressed. By looking at the literature I get the feeling that the authors are not very familiar with the basic statistical copula literature [*Nelsen*, 2006; *Joe*, 2015], which may have prevented them from coming up with a comprehensive list. While I am in favor of a piece addressing such pitfalls, I rather see this as a technical note, a review, or a commentary than an independent piece of research because it does in my point of view not present novel concepts, ideas, tools or data. In addition, I think that such a manuscript should properly review and cite the statistical copula literature and acknowledge previous practical guides for copula application in hydrology e.g. by [*Genest and Favre*, 2007]. In addition, it should not create the wrong sense that dependence=correlation because dependence is a wider term including other dependence properties such as symmetry or tail dependence [*Joe*, 2015]. Even though I do not see this manuscript as a paper in the scope of HESS, I provide some suggestions of how to improve it because I think it could be published as a review/commentary in another hydro-meteorological journal after major modifications.

**Specific comments**

**Title:** I think that the title is too general. The study only reviews manuscripts related to P-T copula analyses and some of the pitfalls described are very specific to that pair of variables (e.g. ties, zero values). I would rephrase it to something like: 'Copulas for joint precipitation-temperature studies – a practical note on common misconceptions and pitfalls'.

**Abstract:** goal, methods, and outcome of study are clearly described. I would probably summarize the different pitfalls identified to summarize the conclusions.

**Introduction:** The study content is generally well introduced. I would personally jump into P-T analyses a bit more directly by removing the first paragraph because it rises the expectation that different hydro-climatic variables are addressed in the manuscript, which is not the case. Instead, I would extend the section on where previous studies looking at hydro-climatic variables are introduced (l. 89-92) because I think this short section does not do the existing literature justice. I would in particular better introduce the study by [*Genest and Favre*, 2007] that describes 'the various steps involved in investigating the dependence between two random variables and in modeling it using copulas' and illustrate these steps on a hydrological example. I would also highlight what exactly is the benefit of your study compared to this previous one, which had a very similar goal.

**Step by step copulas:** This section in my opinion needs a more solid theoretical/statistical basis. Proper citations to the statistical literature should be provided for all equations and statements. All variables should be introduced properly and used consistently. I would furthermore expect a discussion of the following points:

(1) Two-dimensional copulas are not popular in hydro-climatology because people are necessarily interested in only two variables but rather because they are easier to apply and visualize than higher dimensional copulas.

(2) I would add a short section about when to use empirical rather than theoretical copulas and vice versa.

(3) I would mention that copulas model the form and intensity of dependence between variables. The form can be represented by the choice of the copula function while the copula parameter describes the intensity.

(4) I would introduce the notion of dependence and clearly state that correlation only describes one particular part of a dependence structure which also comprises tail dependence [*Poulin et al.*, 2007] or symmetry characteristics. I would recommend having a look at Chapter 2 in [*Joe*, 2015]. These additional characteristics are very important for choosing a suitable copula form.

(5) The particular copulas introduced seem a bit random. Why are extreme value copulas not introduced? They are very important when looking at joint P-T extremes. At least, it should be specified how you determined 'the five most widely used copulas' (l.143).

(6) Some Archimedean copulas have more than one parameter (l.166) and Archimedean copulas have the disadvantage that the same degree of dependence is assumed for all pairs of variables.

(7) Elliptical copulas have the advantage that they can handle the same degree of dependence for different variables pairs but they have symmetric dependence structures which may be a disadvantage in some cases [*Favre et al.*, 2018].

(8) I would treat parameter estimation methods separately and mention why maximum likelihood in some cases can be computationally very expensive and may be replaced by pseudo-maximum likelihood estimation (l. 194) [*Han and De Oliveira*, 2019].

(9) I think equation 16 is wrong. Where does it come from?

(10) A goodness-of-fit test never 'accepts' a hypothesis but rather 'rejects' it. 'non-rejection' does not imply 'acceptance' (l. 238).

(11) It is new to me that NSE and RMSE can be used as copula evaluation metrics (l. 242-245). NSE is used to evaluate time series rather than distributions. I do not see the link to the

dependence structure (except that correlation is evaluated as part of NSE) and neither is the statement underlined by a reference.

**Common issues, misconceptions and pitfalls:** I would move the methods description (l. 248-264) to some methods section. I would also describe how the 'six aspects' investigated (l.255-264) were determined. As mentioned in my general remarks, I would also look at whether authors characterized the nature of dependence (e.g. by looking at rank scatterplots or by computing different dependence metrics including tail dependence) and I would look at whether they performed a proper goodness-of-fit test [*Genest et al.*, 2009]. Furthermore, I would suggest to illustrate the different concepts on your case study example in a separate section called 'Application'.

**Spatio-temporal scale:** It remains unclear to me why exactly this matters unless you wanted to model spatial dependencies. I would introduce the case study in the newly created methods section as suggested above (l. 278-280).

**Correlation:** I would call this section 'Dependence' and discuss dependence aspects going beyond correlation as assuming dependence=correlation is a pitfall in itself (see also my earlier comments). How should correlation be independent of the selected sample? (l. 288-289). By 'generate copula co-dependence structure' do you mean 'fitting a copula structure'?

**Stationarity of correlation:** I would not say that the detection of non-stationarity per se precludes a copula analysis. However, it requires the use of a proper non-stationary model [e.g. *Ahn and Palmer*, 2016]. I do not see the value of the resampling experiment (l. 319-327). Why should this be useful to detect non-stationarity? Why not just test how mean and variance change over time?

**Autocorrelation:** 'time series are dependent on a delayed copy of themselves'?

**Correlation of data with the same rank:** use the term 'ties'. I would remove subsection 3.5.1 because there is just one subsection at that hierarchy level.

**Selecting suitable copula families:** I would remove the NSE and RMSE part (l. 399-400).

**Decision support framework for applying copulas:** would add 'to jointly model P and T' because some of the points are very specific (particularly the one on ties). I would re-order the different steps and put pre-treatment steps such as removal of autocorrelation (2), testing for non-stationarity (3) and ties (4) before dependence assessment (2) and copula fitting (6). I would also include two additional steps: (x) visual inspection of dependence structure and (x) goodness-of-fit testing. My new suggested order is the following: (1) scale (if this is even important), (2) removal of autocorrelation, (3) removal of ties, (4) testing for non-stationarity, (5) visual dependence assessment, (6) computation of dependence metrics, (7) copula fitting, (8) goodness-of-fit tests. Maybe you could even have two main parts called (A) pre-treatment and (B) copula analysis.

**Concluding remarks:** I would discuss which parts of the decision framework are transferable to other variable pairs and which ones are specific to P-T analyses.

**Structure and language:** The manuscript generally has a nice flow and would profit from some editing.

**References:** Some additions from the statistical literature required as specified above.

Figures: In general, I would recommend the use of subplot labels (a, b, c) to facilitate referencing.
Figure 2: What do these turquoise bars on the left and lower part of the figure to the right mean?
Figure 3: would remove the grey borders in the figures to the right (point clouds).
Figure 4: would use distinct colors in the different subplots (different shades of turquoise are used

for lakes).

Figure 5: As mentioned above, I do not see the value of the analysis presented in 5b.

Figure 6: remove random black borders and increase legend (one should be enough).

Figure 7: Would recommend to add isolines to the scatterplots in 6b.

Figure 8: Would recommend to restructure figure according to the steps order suggested above.

**Minor points**

- I am less familiar with the term 'co-dependence' than 'interdependence'. Evtl. reword? (e.g. l. 12).
- l. 11-13. I would restructure the sentence and start with the subject 'Several multivariate analysis approaches have….. to account for precipitation….
- L. 55: I would talk about 'joint' instead of 'compound'.
- The use of commas could be improved, e.g. l. 70 'At the annual resolution,…' or l. 73: 'However,…'
- L. 136: 'can be' instead of 'needs to be'
- L.139: 'provide' instead of 'provides'

**References used in this review**

Ahn, K. H., and R. N. Palmer (2016), Use of a nonstationary copula to predict future bivariate low flow frequency in the Connecticut river basin, *Hydrol. Process.*, *30*(19), 3518–3532, doi:10.1002/hyp.10876.

Favre, A.-C., J.-F. Quessy, and M.-H. Toupin (2018), The new family of Fisher copulas to model upper tail dependence and radial asymmetry: properties and application to high-dimensional rainfall data, *Environmetrics*, *29*(3), 1–17, doi:10.1002/env.2494.

Genest, C., and A.-C. Favre (2007), Everything you always wanted to know about copula modeling but were afraid to ask, *J. Hydrol. Eng.*, *12*(4), 347–367, doi:10.1061/(ASCE)1084-0699(2007)12:4(347).

Genest, C., B. Rémillard, and D. Beaudoin (2009), Goodness-of-fit tests for copulas: A review and a power study, *Insur. Math. Econ.*, *44*, 199–213, doi:10.1016/j.insmatheco.2007.10.005.

Han, Z., and V. De Oliveira (2019), Maximum likelihood estimation of Gaussian copula models for geostatistical count data, *Commun. Stat. Simul. Comput.*, 1–26, doi:10.1080/03610918.2018.1508705.

Joe, H. (2015), *Dependence modeling with copulas*, CRC Press. Taylor & Francis Group, Boca Raton.

Nelsen, R. B. (2006), *An introduction to copulas*, 2nd ed., Springer Science & Business Media, New York.

Poulin, A., D. Huard, A.-C. Favre, and S. Pugin (2007), Importance of tail dependence in bivariate frequency analysis, *J. Hydrol. Eng.*, *12*(4), doi:10.1061/(ASCE)1084-0699(2007)12:4(394).

---

## Short Comment (SC2) · 28 Aug 2020

**Introduction**

In their manuscript, Tootoonchi et al. (2020) discuss common pitfalls when applying copulas in hydroclimatic research. As this mathematical tool is only gaining in popularity, this is an interesting perspective and a subject worth discussing. However, some concepts could be more finetuned, and I hope my comments will help the authors to do so.

**Specific comments**

L. 41 Here you use evapotranspiration, but you used evaporation earlier. Is the use of both terms a conscious choice? If not, the recent preprint by Miralles et al. (2020), gives some insight in the discussion on the word choice.

L. 47-62: Although you mention this slightly (Hao et al. (2018), 'compound context'), I think this paragraph could benefit from a better overview of the growing compound extremes literature, by referring to e.g. Leonard et al. (2014), Zscheischler et al. (2018) or Zscheischler et al. (2020)

L. 86: I think it would be better to cite the textbook by Nelsen (2006) instead of Schweizer (1991), as the first version of that textbook certainly invigorated the study and application of copulas, especially as you do not cite it anywhere else.

L. 89: Instead of Salvadori and De Michele (2010), I think it would be more relevant to cite e.g. Salvadori and De Michele (2004), although there are even earlier papers on the use of copulas in hydrometeorology

L. 98: Given that more and more papers use the term 'bias adjustment', it could be interesting to consider this term as well, as it more clearly states that the biases are and cannot fully be corrected. See e.g. Vrac (2018), Räty (2018), Zscheischler et al. (2019).

L. 101: Only referring to Räty et al. (2018) understates, in my opinion, the discussion on the use of multivariate bias-adjusting methods. For example, Meyer et al. (2019) and Zscheischler et al. (2019) have different conclusions than Räty et al. (2018), and François et al. (2020) clearly show where multivariate bias adjustment methods work and do not work.

L. 104: Schölzel and Friederichs (2008) also serves as a thorough introduction to copulas for hydroclimatic research and is worth citing as well

L. 125: For the copula density definition, it seems more logical to cite a textbook. For example, I easily found this definition in Joe (2014)

L. 136: Although Genest and Favre (2007) certainly clearly proposed how the empirical copula could be practically used, I think you should also cite Deheuvels (1979), as the empirical copula was originally proposed in this paper.

L. 143-148: It surprises me that it is not discussed how often the same few copulas are chosen in studies. This could be a potential pitfall as well. Although it is probably less of a problem for the combination of precipitation and temperature, it could be worth touching upon, especially as you cite Sadegh et al. (2017), wherein this was also discussed (hence the use of many more than the standard copulas in the toolbox presented in that paper).

L. 144: This is a mistake often made, but Archimedean and Elliptical copulas refer to classes instead of families. Gumbel, Clayton... are copula families. See e.g. Nelsen (2006).

L. 163: It would be interesting to see some references on the use of Archimedean copulas for the modeling of extreme events.

L. 168: This sentence seems to imply that there are only three different types of Archimedean copulas, although there are many more (see e.g. Nelsen (2006)).

L. 181: It would be interesting to see some examples of where exactly elliptical copulas are used.

L. 225- 247: On what is this selection based? As you intend to give a good overview of the pitfalls, I would also expect a good overview on the fitting and goodness-of-fit. See e.g. Genest and Favre (2007) (which you already cite).

L. 230: If a statistics is 'widely used', I'm interested in more than one reference, so I can compare the use in different sources. However, your opinion on this may differ.

L. 305-327: I think you could expand this discussion in some regards. First, you only speak about the (non)stationarity of the correlation. However, what about the stationarity of the marginals? These could (and are changing) as well, what could influence the

copula choice. Although this is linked with the correlation strength, I think it deserves a separate discussion. Second, although you already cite a few papers, stationarity is a heavily discussed subject. Some other interesting articles are e.g. Milly et al. (2008), Koutsoyiannis and Montanari (2015) and Serinaldi and Kilsby (2015). Third, part of this discussion is based on the statistical definition of stationarity, which you do not mention. Yet, this could be interesting, and the arguments in Koutsoyiannis and Montanari (2015) are based on this definition. Fourth, it would be interesting to discuss how, if problems with stationarity would arise, this could be dealt with. See e.g. the textbook by AghaKouchak et al. (2012).

L. 348-370. Some older papers also deal with the ranking problem, see e.g. Salvadori and De Michele (2006, 2007) and Vandenberghe et al. (2010). Besides, it is also important to consider that ties do not only occur because of dry days, but also because of measurement imprecision. Depending on the measurement error/discretization and time series length, time series can also contain several ties on wet days.

L. 423-440. Although I admit it is not the essence of the paper, I would be interested in a short discussion/acknowledgement of two aspects. First, would some of the pitfalls become more important when considering other hydroclimatic variables? Second, how would these pitfalls propagate when considering multivariate copula constructions? You for example cite Allen et al. (2017), in which vine copulas are used, which is an important tool for multivariate copula construction.

Technical comments

L. 161: I could not retrieve Hofert (2008) in the references

L. 166: 'is defined on (0 1]'. Is this correct?

L. 209: Should this instead be 'given a specific value of a second variable'

References

AghaKouchak et al. (2012): Extremes in a changing climate

Deheuvels (1979): La fonction de dépendance empirique et ses propriétés. Un test non parametrique d'indépendance. Bulletin de la classe des sciences, Academie Royale de Belgique

François et al. (2020): Multivariate bias corrections of climate simulations: Which benefits for which losses? Earth System Dynamics, https://doi.org /10.5194/esd-11-537-2020

Joe (2014): Dependence modelling with copulas

Koutsoyiannis and Montanari (2015): Negligent killing of scientific concepts: the stationarity case. Hydrological Sciences Journal, https://doi.org/10.1080/02626667.2014.959959

Leonard et al. (2014): A compound event framework for understanding extreme impacts. Wiley Interdisciplinary Reviews: Climate Change, https://doi.org/10.1002/wcc.252

Meyer et al. (2019): Effects of univariate and multivariate bias correction on hydrological impact projections in alpine catchments. Hydrology and Earth System Sciences, https://doi.org/10.5194/hess-23-1339-2019

Milly et al. (2008): Stationarity is dead: Whither water management? Science, https://doi.org/10.1126/science.1151915

Miralles et al. (2020): On the use of the term 'Evapotranspiration'. Earth and Space Science Open Archive, https://doi.org/10.1002/essoar.10503229.1

Nelsen (2006): An introduction to Copulas, 2nd edition.

Salvadori and De Michele (2004): Frequency analysis via copula: theoretical aspects and applications to hydrological events. Water Resources Research, https://doi.org/10.1029/2004WR003133

Salvadori and De Michele (2006): Statistical characterization of temporal structure of

storms, https://doi.org/10.1016/j.advwatres.2005.07.013

Salvadori and De Michele (2007): On the use of copulas in hydrology: theory and practice. Journal of Hydrologic Engineering, https://doi.org/10.1061/(ASCE)1084-0699(2007)12:4(369)

Schölzel and Friederichs (2008): Multivariate non-normally distributed random variables in climate research-introduction to the copula approach. Nonlinear Processes in Geophysics, https://doi.org/10.5194/npg-15-761-2008

Serinaldi and Kilsby (2015): Stationarity is undead: Uncertainty dominates the distribution of extremes. Advances in Water Resources, https://doi.org/10.1016/j.advwatres.2014.12.013

Vandenberghe et al. (2010): Fitting bivariate copulas to the dependence structure between storm characteristics: A detailed analysis based on 105 year 10 min rainfall. Water Resources Research, https://doi.org/10.1029/2009WR007857

Vrac (2018): Multivariate bias adjustment of high-dimensional climate simulations: the Rank Resampling for Distributions and Dependences (R2D2) bias correction. Hydrology and Earth System Sciences, https:// doi.org/10.5194/hess-22-3175-2018

Zscheischler et al. (2018): Future climate risk from compound events. Nature Climate Change, https://doi.org/10.1038/s41558-018-0156-3

Zscheischler et al. (2019): The effect of univariate bias adjustment on multivariate hazard estimates. Earth System Dynamics, https://doi.org/ 10.5194/esd-10-31-2019

Zscheischler et al. (2020): A typology of compound weather and climate events, Nature Reviews Earth & Environment, https://doi.org/10.1038/s43017-020-0060-z

---

## Referee Comment (RC2) · Geoff Pegram (Referee) · 16 Sep 2020

What a pleasure it was to review this article. This is possibly the best Hydrometeorological paper that I have read in the last few years and is a must-read in this genre. It is targeted at authors involved with, or starting off to work with, copulas in time series. The difficulty that presents itself when analysing time series characterised by serial correlation, is that that for analysis, modelling or forecasting, the leading question is: 'how do I get a handle on this problem?' The beauty of the paper is that it a distillation of ideas into a rubric for preparing an analysis of one or more time series, to be finished off with a flow chart for guidance.

[Figure]

It is an important reminder and guide for time series analysts, and is not only tutorial, but is wisely, simply, and authoritatively compiled. In my judgement this should be published once some small issues have been dealt with. For example, the authors should attend to some cosmetic suggestions to fix the few spelling and grammatical errors, as well as embellishments in the figure and table captions to make them more readable. Again, stylistically, it would improve the readability if you either add a space between all paragraphs or indent the leading line. Also, I could not find 'saturation water vapor mixing ratio' this paper. My more pertinent remarks follow.

In section 3, line 326, you state: 'We adopted a copula framework on July, because it has significant correlation at both daily and monthly resolutions (Fig. 5(a)-(b)).'

Did you try lagging the daily precipitation and streamflow data? Surely the delay depends on the size of catchment.

Line 338: 'It is important to consider the degree of autocorrelation in the studied data.' Without pushing my co-authorship of a relevant paper, to check the effect of autocorrelation, you might like to look at: Sugimoto, Takayuki, András Bárdossy, Geoffrey G.S. Pegram, and Johannes Cullmann (2016), Investigation of hydrological time series using copulas for detecting catchment characteristics and anthropogenic impacts, Hydrol. Earth Syst. Sci., 20, 2705 -2720, doi:10.5194/hess-20-2705-2016.

In Figure 6: What is the spread of the confidence intervals - 95%? In (d) it looks like 100%

In Figure 8: that's a very helpful flow-chart - especially the '!!'

In Table 2: please define the symbols in the caption to help the reader: ERMS, ENS, Sn and Dn

There are also a few minor suggestions that I have made for alteration, so I am uploading my marked-up copy of the paper with this review for the authors' information. Well done!

Geoff Pegram

Please also note the supplement to this comment:
https://hess.copernicus.org/preprints/hess-2020-306/hess-2020-306-RC2-
supplement.pdf

———————————————————
306, 2020.

**Supplement:**

**Copulas for hydroclimatic applications - A practical note on common misconceptions and pitfalls**

Faranak Tootoonchi1, Jan Olaf Haerter2, Olle Räty3, Thomas Grabs1, Mojtaba Sadegh4, Claudia Teutschbein1

5 1 Department of Earth Sciences, Uppsala University, Uppsala, Sweden

2 Niels Bohr Institute, University of Copenhagen, Copenhagen, Denmark

3 Finnish Meteorological Institute, Helsinki, Finland

4 Department of Civil Engineering, Boise State University, Boise, USA

Correspondence to: Faranak Tootoonchi (faranak.tootoonchi@geo.uu.se)

- 10 Abstract. For most hydroclimatic applications, precipitation and temperature are of particular interest as they strongly affect the water cycle, can easily be measured and are often readily available from many meteorological stations worldwide. To account for precipitation and temperature variability, their co-dependence and their correlation, several multivariate analysis methods have been adopted in the hydroclimatic literature in recent years. In line with the steadily rising number of publications on this topic, the notion of copula-based probability distribution has also attracted tremendous interest to deal with the
- 15 complexity of compound events in the multidimensional context. A copula is a function that connects a multivariate distribution to its one-dimensional margins, which allows for a joint distribution of random variables with great flexibility for the marginal distribution. However, there seems to be a lack of comprehensive understanding of the fundamental requirements of the copula concept such as the strength and significance of correlation between variables, autocorrelation effects and the choice of representative copula families, which potentially compromises the robustness of projections of future environmental
- 20 processes and natural hazards. Therefore, by combining a systematic literature review with a specific hydroclimatic case study in Sweden, we illustrate a practical approach to copula-based modeling, which (1) provides end-users with an overview of necessary requirements, statistical assumptions and consequential limitations of copulas, (2) highlights possible pitfalls and misconceptions, and (3) offers a decision support framework for the application of copulas to support researchers and decision makers in addressing climatological hazards and sustainable development, thereby demystifying what is currently an area of
- 25 great confusion.

**1. Introduction**

30

Water touches every aspect of our lives, from public health to safety, to the foundation of our economy. Alterations to the hydroclimatic drivers of the water cycle form a potential threat, as they influence socioeconomic, ecological and climate systems (Vogel et al., 2015). In this paper, we adopt the definition of hydroclimatology as the "study of the influence of climate upon the waters of the land" originally proposed by Walter Langbein in the 1970's and later further expanded to include hydrometeorology (i.e., the land-atmosphere interface) as well as the surface and near-surface water processes of evaporation, runoff, groundwater recharge and interception (Wendland, 1987). As such, hydroclimatology encompasses a large number of

35

**stylistically, it would improve the readability if you either add a space between all paragraphs, or indent the leading line**

Hydrology and S

Earth System

Sciences

Discussions

[revised manuscript text omitted]

340 of those addressed it by removing it or by bootstrapping.

For the Vattholma catchment, the autocorrelation function (ACF) for precipitation (Fig. 6(a)) and temperature (Fig. 6(b)) in July indicates some autocorrelated behavior. In order to account for autocorrelation, Laux et al. (2011) suggested using a firstto check the effect of autocorrelation you might like to look at: Sugimoto, Takayuki, András Bárdossy, Geoffrey G.S. Pegram, and Johannes Cullmann (2016), Investigation of hydrological time series using copulas for detecting catchment characteristics and anthropogenic impacts, *Hydrol. Earth Syst. Sci.*, 20, 2705 -2720, doi:10.5194/hess-20-2705-2016.

---

## Short Comment (SC3) · 28 Sep 2020

In agreement with RC1, I also "get the feeling that the authors are not very familiar with the basic statistical copula literature". However, I also get the feeling that the Authors seem to be not even familiar with some basic statistical concepts as well as literature dealing with applications of copulas to hydrological variables. In this respect, it is quite ironic or paradoxical that a paper discussing "misconceptions" endorses and re-proposes "misconceptions"! I also think that this type of papers should be written/supervised by people with more experience in the field; I mean names like Favre, Genest, Salvadori, De Michele, Bardossy, and some others... almost certainly, this is

not a task for people with limited experience, in my opinion.

As I recognize that the above statements can appear harsh, please, let me discuss only few points to support my opinion.

L70: Before quoting a paper, it is better to be sure about its content. For example, the Authors state "At the annual resolution AghaKouchak et al. (2014) found a significant negative correlation between anomalies of temperature and precipitation over California". However, those data show zero Kendall correlation, as can be seen by reading a bit more carefully the cited Serinaldi (2016), who re-analyzed the same data, and showed that the sample size is not enough to make conclusions on the actual dependence structure.

L88: "Since the early 2000's, copula methods have been adopted in hydrological modeling, which was triggered by the study of Salvadori and De Michele (2010)." Another discussant suggested Favre et al. (2004) and Salvadori and De Michele (2004) as the first applications of copulas in hydrological domain. Well, the first paper applying copulas in the hydrological context is De Michele and Salvadori (2003), whose 15th anniversary was also celebrated by a special issue in Water journal (https://www.mdpi.com/journal/water/special_issues/copulas_hydrology). Knowing history can help.

L103: "only a few papers also relate to practical challenges encountered in hydroclimatic research"... Often, my papers have been criticized as they are too critical when discussing the problems/challenges of performing inference (including copula inference) on hydroclimatic data; so, it is quite funny to discover that people have criticized me for nothing.

L109: "Accordingly, this paper aims at filling this gap and serves as an overview of the state of the art of using copulas in hydroclimatology for practitioners interested in adopting this method for their research." As discussed below, this paper is far from filling any gap, and surely does not report the state of the art in any respect.

L140: "The result would be the joint CDF of uniformly distributed marginal"?? Perhaps, "the joint CDF of uniformly distributed random variables". Before talking about misconceptions, it can be good to familiarize with the nomenclature and the meaning of the words used.

L145: this simulation procedure yields fitted joint distributions characterized by different (thoeretical) Pearson correlations. The correct procedure to show the difference of the tail behavior for identical marginals and Pearson/Kendall correlations is to simulate samples from distributions with specified copula, marginals and Pearson/Kendall correlations. However, since the Pearson correlation depends on both copula and marginals (as stressed by another discussant), and generating samples with the same (theoretical) Pearson correlation can be a bit tricky, my feeling is that the Authors opted for this shortcut, which is however theoretically incorrect and numerically imprecise.

Section 2.3: The Authors merge the expression of the population Pearson correlation with the sampling estimators of the Kendall and Spearman correlations. This generates not only confusion but also a subsequent mistake concerning how to account for zeros and more generally for statistical ties (see below). When presenting dependence measures, it can be better to discuss both theoretical expressions and finite sample estimators, just to avoid "misconceptions".

L168: "Aloui, Ben Aïssa, and Nguyen (2013) distinguished between three different types of Archimedean copulas" I'm sure there are better references for this, and "distinguished" seems to me inappropriate (perhaps "discussed" or "focused on").

L179: "After estimating $\theta$ based on Kendall's $\tau$ (Table 1), the copulas can be computed using their respective formulas." For a paper that should discuss the "state of the art", only mentioning the moment-like estimation based on Kendall $\tau$, which is not even applicable to multi-parameter bivariate copulas, seems to me insufficient. What about max-likelihood (with all its flavors: exact ML, max pseudo-like, etc.), max-entropy, Bayesian inference, etc.?

[Figure]

Sec. 2.3.2: please use $\Phi_R$ to denote an elliptical multivariate distribution (as per Fig. 3).

L203: "Embrechts, Mcneil, and Strauman (1999) estimated the Elliptical copula parameter from the rank-based Kendall's $\tau$ or Spearman's $S$.": this is valid only for elliptical copulas, such as Gaussian, that are only characterized by pairwise correlation matrices. For families such as the Student copula, we need to estimate additional parameters (e.g. the degrees of freedom), and therefore other estimators are required (if we want to report "the state of the art").

L237: "The null hypothesis of the test is the acceptance of the parametric copula. For a particular copula, the p-value is sufficient to determine the acceptance or rejection of the null hypothesis with the significance level of $\alpha$, but in a group of different acceptable copulas, the best alternative is the one with the smallest $S_n$ or $D_n$ and the greatest p-value (Madadgar and Moradkhani, 2014)." This sentence is incorrect. Leaving aside the meaning (or lack of meaning) of the hybrid of Fisher-Neyman-Pearson hypothesis testing framework (see Wasserstein et al. (2019) and references therein along with ASA recommendations), goodness-of-fit tests are never confirmatory; they can only conclude that "the null cannot be rejected", meaning that there is not enough information to exclude the null hypothesis. The last part of the of Authors' statement describes another widespread misconception. Indeed, test statistics and p-values cannot be used to rank the models. Why? Because the model parameters are estimated on the data, and therefore the KS and CvM tests are no longer distribution free, and then those p-values correspond to different quantiles of different distributions of the test statistics. This is the reason why the null distribution of GRB goodness-of-fit test is computed by MC simulation, which must be performed every time we consider a different data set. Instead of reporting suggestions described by hydrologists "playing" with statistics, I suggest reading statistics written by statisticians such as D'Agostino and Stephens (1986)... or hydrologically oriented but theoretically grounded papers such as Laio (2004), for a gentle discussion.

L242: Using an ensemble of performance measures makes sense if these measures highlight different "fitting" aspects. However, these indexes are often selected quite randomly, as in this case, overlooking their redundancy; indeed, Nash-Sutcliffe is nothing but a similarity measure corresponding to the (R)MSE (see e.g. Hyndman and Koehler (2006), Jachner et al. (2007), Dawson et al. (2007), and Reusser et al. (2009) for a discussion on a more appropriate use of performance metrics). Moreover, selecting the model with the lowest/better metrics is also questionable especially in the case of (usually) small samples, as these metrics are affected by uncertainty, and the model rank can change by changing the sample. This problem is well known in the field of "information criteria" (AIC, BIC, etc.), where the selection either relies on the significance of the differences between two models (in terms of model evidence) or it is somewhat avoided by using model averaging.

Sec.3 and Fig. 8: In my opinion, most of the supposed pitfalls and misconceptions listed in Sec. 3 and summarized in the flow chart in Fig. 8 result from some Authors' misconceptions or rather superficial approach to the topic. Firstly, copulas are general models that can be used for data at any spatio-temporal scale. Even if data at some scales can generally be more or less serially correlated, for instance, this does not prevent the use of copulas, taking for granted that serial correlation should be accounted for in some way. However, the problem should be considered case-by-case rather than ascribed to spatio-temporal scales. Moreover, serial correlation or other properties depend on the variables at hand. Furthermore, checking for the significance of the cross-correlation of two (or more) variables is a false problem. Indeed, we use copulas to build joint distributions, thus meaning that we need the joint distributions, independently of the correlation value. When the correlation is close to zero, this simply means that the product copula is a feasible option, and this copula is often a special case of other copulas. As mentioned by another discussant, it seems to me that the Authors confuse correlation and dependence structure. Zero correlation does not mean that the joint probability is zero or does not exist, or it is not of interest; it means that the joint probability can reasonably be described by the product of marginals or, the same,

by the product copula. In this respect, the sentence "...there are months in which the correlation is not significant at the 5% level, thus cannot be considered as reliable correlation between variables." makes little sense. Zero correlation is what is, it is not less/more "reliable" than 0.2, 0.5, or 0.9. Reliability depends at most on the sample size used to estimate the correlation values. We can use copulas for every month in Fig. 5 if we need the joint distribution of every month. Selection of June and July is artificial, not required, and not justified neither empirically nor theoretically.

Sec. 3.3: Another discussants already made comments on the stationarity issue. I would only like to stress that the sentence "when the correlation is highly sensitive to the selected time period, it is an indication of a non-stationary behaviour" makes little sense. Non-stationarity requires a (known) law of evolution, while sub-sample fluctuations do not indicate any non-stationarity per se. The Vattholma example and corresponding numerical experiment and interpretation is also meaningless. Indeed, that bootstrap experiment does not (and cannot) reveal non-stationarity; it simply shows the sampling variability for 10-year samples under stationarity! Why? Because the bootstrap experiment (selection of ten randomly chosen years) is designed to destroy whatever supposed time evolution. Fig. 5 only shows the seasonality of the correlation and its sampling uncertainty for 10-year spanning samples.

Sec. 3.4 has missed the key works of Andrew Patton providing the theoretical basis and conditions required to apply the so-called conditional or dynamic copulas, i.e. the models applied in the cited references. The sentence "In contrast, other papers argued that copulas are still applicable in case of only low-degree (removable) auto-correlation in the time series" denotes once again some lack of familiarity with the existing literature. Indeed, these models are widely applied in econometrics, involving complex and often strongly persistent processes (see also Serinaldi and Kilsby (2017) for an example on hydrological data). Relying exclusively on hydrological literature to get information about statistical tools is never a good idea; based on my experience, hydrology is one of the disciplines with the most superficial (amateur) use of statistics.

Sec. 3.5: "Correlation for data with the same rank (dry days)". Data with the same rank are called statistical ties, and unlike stated in this section they are not a big problem and do not require any rainfall threshold, post-processing or such. What is needed is only a decent literature review revealing that (i) estimators of e.g. Kendall correlation accounting for ties and zero-inflation already exist and do not require any pre/post processing (check Kendall, Gibbons and others' works), and (ii) joint distributions of two variables with discrete-continuous and continuous marginals, respectively, are special cases of the models described for instance by Shimizu (1993) and Herr and Krzysztofowicz (2005), and formalized in terms of copulas by Serinaldi (2009). As mentioned above, distinguishing between population and sampling version of a given statistic can help avoiding mistakes.

L379: "Since a significant correlation is the basis for applying copulas". I do not think so: the need for joint distributions with specified dependence structure and marginals is the basis for applying copula. If such a copula is the product copula in some cases, it does not matter very much.

L389: "We here demonstrate how such performance measures can be applied practically"; as mentioned above, Sec. 3.6 demonstrates the opposite, i.e. how not to apply these indexes.

- $S_n$ or $D_n$ are associated to goodness-of-fit tests that have only dichotomous outcome (rejection or not rejection);

- they (and p-values) cannot be used to rank the models (as explained above);

- the range of the four indexes are different and there is no empirical or theoretical support for the statement "Here, evaluation of copulas by $E_{RMS}$ or $E_{NS}$ did not reveal the weak performance of Clayton copula".

Based on Table 2 and considering the similarity of Gaussian and Student copulas (as p-values are not reported), the interpretation should be as follows:

- CvM and KS tests probably say that those two models cannot be rejected, while Clayton can be (p-values should be shown to confirm this);

- $E_{RMS}$ or $E_{NS}$ do not allow any conclusion without complementing them with uncertainty assessment, which quantify the sampling fluctuations of these metrics (and the significance of their differences).

In my opinion, this paper does not provide "an overview of the state of the art of using copulas in hydroclimatology", but something like the opposite. Based on a very superficial literature review (which neglects theoretical literature), and an apparent lack of familiarity with the topic, this manuscript is also quite superficial itself, iterates some misuses of statistical tool (which are widespread in the hydrological literature), and does not provide a good service to a community that already suffers from confusion when coming to applied statistics.

Based on my experience, most of the misconceptions concerning copulas, and more generally applied statistics, are related to hydrolgists' statistical background, which is on average much more limited than that of people working on other fields, such as economics, biology, medicine, etc., where statistical analysis is routinely performed by (or with the help of) professional statisticians, or people with much more solid statistical background. I also think that the sampling and model uncertainty mentioned by the Authors in their conclusions is one of the most important aspects to draw meaningful conclusions. In most of the submitted or published literature on these topics, data are not enough to draw any definite conclusion, and statistics is somewhat (ab)used to overcome this lack of information, with the incorrect hope that it can give that certainty (or rigor) that the data and meta-data cannot provide. Concerning the uncertainty in copula inference it could be fair mentioning the contributions of Serinaldi (2013), Dung et al. (2015), and Zhang et al. (2015). Once uncertainty is accounted for, copula inference often reveals that the discrimination among different models and/or preliminary assumptions is very difficult, if not impossible without additional information.

My final (not requested) suggestion for the Authors is to reconsider their approach to this topic, going beyond the concepts that can be deduced from few tens of hydrological papers, which may be not even the best ones in terms of quality. As mentioned above, such a kind of papers require a wider/deeper familiarity with theory, applications, literature (from different fields), etc.

Sincerely

Francesco Serinaldi

References

D'Agostino RB, Stephens MA (1986). Goodness-of-fit-techniques (Vol. 68). CRC press

Dawson, C. W., Abrahart, R. J., See, L. M. (2007). HydroTest: a web-based toolbox of evaluation metrics for the standardised assessment of hydrological forecasts. Environmental Modelling Software, 22(7), 1034-1052

De Michele, C., and G. Salvadori (2003) A Generalized Pareto intensity-duration model of storm rainfall exploiting 2-Copulas, J. Geophys. Res., 108(D2), 4067, doi:10.1029/2002JD002534

Dung NV, Merz B, Bárdossy A, Apel H (2015) Handling uncertainty in bivariate quantile estimation - an application to flood hazard analysis in the Mekong Delta. J Hydrol 527:704–717

Herr HD, Krzysztofowicz R (2005) Generic probability distribution of rainfall in space: the bivariate model. J Hydrol 306:234–263

Hyndman, R. J., Koehler, A. B. (2006). Another look at measures of forecast accuracy. International journal of forecasting, 22(4), 679-688

Jachner, S., Van den Boogaart, G., Petzoldt, T. (2007). Statistical methods for the qualitative assessment of dynamic models with time delay (R Package qualV). Journal

of Statistical Software, 22(8), 1-30

Laio, F. (2004), Cramer-von Mises and Anderson-Darling goodness of fit tests for extreme value distributions with unknown parameters, Water Resour. Res., 40, W09308, doi:10.1029/2004WR003204.

Patton A.J. (2009) Copula–Based Models for Financial Time Series. In: Mikosch T., Kreiß JP., Davis R., Andersen T. (eds) Handbook of Financial Time Series. Springer, Berlin, Heidelberg

Reusser, D. E., Blume, T., Schaefli, B., Zehe, E. (2009). Analysing the temporal dynamics of model performance for hydrological models. Hydrology and earth system sciences, 13(7), 999-1018

Serinaldi F. (2009) Copula-based mixed models for bivariate rainfall data: an empirical study in regression perspective. Stochastic Environmental Research and Risk Assessment, 23(5), 677–693.

Serinaldi F (2013) An uncertain journey around the tails of multivariate hydrological distributions. Water Resour Res 49(10):6527–6547

Serinaldi F, Kilsby CG (2017) A blueprint for full collective flood risk estimation: demonstration for European river flooding. Risk Analysis, 37(10), 1958-1976

Shimizu K (1993) A bivariate mixed lognormal distribution with an analysis of rainfall data. J Appl Meteorol 32:161–171

Zhang Q, Xiao M, Singh VP (2015) Uncertainty evaluation of copula analysis of hydrological droughts in the East River basin, China. Global Planet Change 129:1–9

Wasserstein RL, Schirm AL, Lazar NA (2019) Moving to a World Beyond "$p < 0.05$", The American Statistician, 73:sup1, 1-19, DOI: 10.1080/00031305.2019.1583913

306, 2020.

---

## Author Comment (AC1) · 2 Nov 2020

**Reply to reviewers for HESS-2020-306 article**

*Title:* Copulas for hydroclimatic applications – A practical note on common misconceptions and pitfalls

*Authors:* Faranak Tootoonchi, Jan Olaf Haerter, Olle Räty, Thomas Grabs, Mojtaba Sadegh, and Claudia Teutschbein

Dear Editor and Reviewers

We would like to thank the editor for providing us the opportunity to discuss the content of this paper with the scientific community. We also would like to thank official reviewers and unofficial reviewers (commenters) for taking the time to examine this manuscript. The latter came as a pleasant surprise as it shows the enormously growing interest in this field and – in a humble tone – interest in this paper.

In this study, we provided end-users with a decision support framework to adopt the copula approach to study the dependence structure between hydroclimatic variables (namely, precipitation and temperature). To back up the proposed framework, we performed a reproducible literature review using the Scopus search engine. We also applied the framework to a case study in Sweden and graphically showed the issues that might arise when this framework is not followed carefully.

We received two sets of official reviews: One detailed feedback from anonymous reviewer #1 which acknowledged positive aspects of the paper (e.g., flow and clarity of the text) but also specifies very useful comments that raise fruitful discussions, which will enable further improvement of this paper.

The main comments from the reviewer were focused on:

*1-Why we submitted this article as a research paper to HESS?*

*2-Did we fairly review and give credit to the statistical papers?*

*3-Did we address most relevant pitfalls?*

We also received another extremely positive review from Geoff Pegrem, quote: *'…What a pleasure it was to review this article. This is possibly the best Hydrometeorological paper that I have read in the last few years and is a must-read in this genre. … It is targeted at authors involved with, or starting off to work with, copulas in time series.'* For which we are extremely thankful. This actually was the motive behind writing this manuscript: To provide the starters in this field with a well-documented, good narrated text to start working with copulas, to be able to later benefit from highly technical papers in this field. We are happy to see that this goal is achieved.

We would like to provide detailed response to reviewers comment here. Afterwards, we also briefly address the points raised by commenters, many of which have also been raised by the reviewers.

**Comment/Response to anonymous reviewer #1**

**General comments**

I agree with the authors that copulas are often used in the fields of hydrology and climatology and that there is a lot of room for improvement in when and how they are applied. In my point of few, the most common 'pitfalls' are that (1) the nature of dependence is often not studied before starting to test various copulas; (2) the dependence structure is often reduced to correlation, (3) no proper goodness-of-fit tests are applied to reject inappropriate copulas. While the authors detect several other pitfalls related to P-T analyses, these important pitfalls are not addressed. By looking at the literature I get the feeling that the authors are not very familiar with the basic statistical copula literature [*Nelsen*, 2006; *Joe*, 2015], which may have prevented them from coming up with a comprehensive list. While I am in favor of a piece addressing such pitfalls, I rather see this as a technical note, a review, or a commentary than an independent piece of research because it does in my point of view not present novel concepts, ideas, tools or data. In addition, I think that such a manuscript should properly review and cite the statistical copula literature and acknowledge previous practical guides for copula application in hydrology e.g. by [*Genest and Favre*, 2007]. In addition, it should not create the wrong sense that dependence=correlation because dependence is a wider term including other dependence properties such as symmetry or tail dependence [*Joe*, 2015]. Even though I do not see this manuscript as a paper in the scope of HESS, I provide some suggestions of how to improve it because I think it could be published as a review/commentary in another hydro-meteorological journal after major modifications.

Again we would like to thank anonymous reviewer #1 for their detailed and thorough feedback. We would like to provide an answer to their three major comments. Which are:

*1-Why we submitted this article as a research paper to HESS?*

*2-Did we fairly review and give credit to the statistical papers?*

*3-Did we address the most relevant pitfalls (pointed by reviewer as numbers 1 to 3)?*

**For the first point:** although we see the potential of this paper as a review paper or a technical note, we argue that we developed a practical decision support framework (= new tool) for the application of copulas to support hydroclimatic researchers, which is illustrated in detail using a particular case study in Sweden (new data). We analysed the data with this approach and discussed the changing nature of dependence within our case study.

To that end, we continue to believe that this manuscript is a perfect fit to be published in HESS, which is aiming "to serve not only the hydrological science community but all earth and life scientists, water engineers, and water managers". Also, because this platform allows for discussions among the scientific community, we believe that our manuscript serves this purpose, because of the clear text that can guide the user in further adoption/adaptation of this framework as it has already been noticed by some researchers. Taking all this into consideration, we think that this manuscript (in an updated version) could potentially attract a wide audience.

**For the second point**, while we extensively used statistical papers such as Genest and Favre (2007) or Genest, C., B. Rémillard, and D. Beaudoin (2009) and referenced them in the

original manuscript, we can see that further credit is due upon these important papers. We will address this in the revised manuscript.

**For the third point,** 1- Yes, that is one of our goals to ensure future researchers comprehensively analyze their data and its dependence structure, before using statistical models such as copula.

2- We do agree that the nature of dependence should not be reduced to correlation and we will further discuss this in the revised manuscript. However, correlation – in general sense, including Spearman rank correlation – is an easy and available test to evaluate the dependence – in its informal definition – between variables. Without putting too much burden on the end-users, correlation tests can give a sense of the data. Further, we did not introduce correlation as a requirement for copula analysis, we simply suggest correlation analysis as a first step for the researchers to understand their data before modeling.

3- We argue that we extensively discussed goodness-of-fit tests (we adopted p-value following the procedure discussed in Genest et al. 2009) both in the methodology section (section 2.5) and later when mentioning copula fitting, and rejection of unsuitable copulas. However, we will try to put more emphasis on this in the revised manuscript.

**Specific comments**

**Title:** I think that the title is too general. The study only reviews manuscripts related to P-T copula analyses and some of the pitfalls described are very specific to that pair of variables (e.g. ties, zero values). I would rephrase it to something like: 'Copulas for joint precipitation-temperature studies – a practical note on common misconceptions and pitfalls'.

We appreciate this comment. The reviewer is right that our case study is focused on precipitation-temperature relationship; however, the nature of the proposed framework and guide to use copulas is general and can be applied to a myriad of variables.

**Abstract:** goal, methods, and outcome of study are clearly described. I would probably summarize the different pitfalls identified to summarize the conclusions.

We appreciate the reviewer's positive comment and will conclude the identified pitfalls.

**Introduction:** The study content is generally well introduced. I would personally jump into P-T analyses a bit more directly by removing the first paragraph because it rises the expectation that different hydro-climatic variables are addressed in the manuscript, which is not the case. Instead, I would extend the section on where previous studies looking at hydro-climatic variables are introduced (l. 89-92) because I think this short section does not do the existing literature justice. I would in particular better introduce the study by [*Genest and Favre*, 2007] that describes 'the various steps involved in investigating the dependence between two random variables and in modeling it using copulas' and illustrate these steps on a hydrological example. I would also highlight what exactly is the benefit of your study compared to this previous one, which had a very similar goal.

We appreciate the reviewer's positive comment. We will address this comment when revising the manuscript.

**Step by step copulas:** This section in my opinion needs a more solid theoretical/statistical basis. Proper citations to the statistical literature should be provided for all equations and

statements. All variables should be introduced properly and used consistently. I would furthermore expect a discussion of the following points:

When revising the manuscript, we will improve this section as suggested.

(1) Two-dimensional copulas are not popular in hydro-climatology because people are necessarily interested in only two variables but rather because they are easier to apply and visualize than higher dimensional copulas.
We will mention this.

(2) I would add a short section about when to use empirical rather than theoretical copulas and vice versa.
We will add this part.

(3) I would mention that copulas model the form and intensity of dependence between variables. The form can be represented by the choice of the copula function while the copula parameter describes the intensity.

We will mention this.

(4) I would introduce the notion of dependence and clearly state that correlation only describes one particular part of a dependence structure which also comprises tail dependence [*Poulin et al.*, 2007] or symmetry characteristics. I would recommend having a look at Chapter 2 in [*Joe*, 2015]. These additional characteristics are very important for choosing a suitable copula form.

We will add this part.

(5) The particular copulas introduced seem a bit random. Why are extreme value copulas not introduced? They are very important when looking at joint P-T extremes. At least, it should be specified how you determined 'the five most widely used copulas'.

We did not include extreme value copulas as we looked at the dependence between complete range of precipitation and temperature. We do agree with the reviewer that justification is needed for the selection of five most widely used copulas. For this, we analyzed copulas that have been used in the studied literature returned from Scopus. This will be reflected in the revised manuscript.

(6) Some Archimedean copulas have more than one parameter and Archimedean copulas have the disadvantage that the same degree of dependence is assumed for all pairs of variables.
We will further assess this.

(7) Elliptical copulas have the advantage that they can handle the same degree of dependence for different variables pairs but they have symmetric dependence structures which may be a disadvantage in some cases [*Favre et al.*, 2018].
We will add this part.

(8) I would treat parameter estimation methods separately and mention why maximum likelihood in some cases can be computationally very expensive and may be replaced by pseudo-maximum likelihood estimation [*Han and De Oliveira*, 2019].
We will include this part.

(9) I think equation 16 is wrong. Where does it come from?

We double-checked and can confirm that this equation is correct. Please refer to eq. 1-6 in Shojaeezadeh et al. 2018

(10) A goodness-of-fit test never 'accepts' a hypothesis but rather 'rejects' it. 'non-rejection' does not imply 'acceptance'.

Agreed.

(11) It is new to me that NSE and RMSE can be used as copula evaluation metrics (l. 242-245). NSE is used to evaluate time series rather than distributions. I do not see the link to the dependence structure (except that correlation is evaluated as part of NSE) and neither is the statement underlined by a reference.

NSE is used herein in accordance with its counterpart Coefficient of Determination that is used for statistical models. This can provide a sense of how close the copula-predicted probabilities are to their empirical counterparts, as opposed to p-values that do not show this. In the studied literature, we found that NSE and RMSE have been used to evaluate copulas. NSE and RMSE are two widely used metrics in hydrological studies. We, however, showed in table 2 that relying solely on these two metrics, might be misleading. We will include more discussion about proper copula evaluation metrics.

**Common issues, misconceptions and pitfalls:** I would move the methods description (l. 248-264) to some methods section. I would also describe how the 'six aspects' investigated (l.255-264) were determined. As mentioned in my general remarks, I would also look at whether authors characterized the nature of dependence (e.g. by looking at rank scatterplots or by computing different dependence metrics including tail dependence) and I would look at whether they performed a proper goodness-of-fit test [*Genest et al.*, 2009]. Furthermore, I would suggest to illustrate the different concepts on your case study example in a separate section called 'Application'.

Thank you for the detailed suggestions. We would like to re-emphasize that we had used p-value as a formal goodness-of-fit measure for copulas in the original manuscript. In revising the manuscript, we will emphasize the importance of rank correlation and rank plots as an appealing approach in determining presence or lack of dependence between variables. When revising the manuscript, we will make some adjustment to the structure.

**Spatio-temporal scale:** It remains unclear to me why exactly this matters unless you wanted to model spatial dependencies. I would introduce the case study in the newly created methods section as suggested above (l. 278-280).

Spatio- temporal scale can be of importance, as the significance of correlation might change depending on the scale (fig 8). This generally speaks to the importance of understanding the physical processes that govern the data generation, before using statistical models for any analysis.

**Correlation:** I would call this section 'Dependence' and discuss dependence aspects going beyond correlation as assuming dependence=correlation is a pitfall in itself (see also my earlier comments).

We agree and will further discuss this.

How should correlation be independent of the selected sample?

We illustrated this in figure 5. We meant here that the correlation must not be merely random (e.g. number of divorces increases as the number of marriages increases, but is marriage a cause for divorce?). We will change this sentence to clarify.

(l. 288-289) By 'generate copula co-dependence structure' do you mean 'fitting a copula structure'?

Yes

**Stationarity of correlation:** I would not say that the detection of non-stationarity per se precludes a copula analysis. However, it requires the use of a proper non-stationary model [e.g. *Ahn and Palmer*, 2016]. I do not see the value of the resampling experiment (l. 319-327). Why should this be useful to detect non-stationarity? Why not just test how mean and variance change over time?

We would like to thank the reviewer by pointing out this very interesting point. We do agree that 'stationarity of the hydroclimatic' can be addressed differently, thus should be discussed more cautiously. In fact, this is a hot topic in hydroclimatic research and we will add more discussion, referencing the papers mentioned by the reviewer.

However, we would like to argue that the motive behind writing this part was not disapproving any use of statistical methods in presence of climate change. Rather by this part and further by illustration in figure 5, we wanted to point out the importance of careful analysis of dependence within a timeframe. Accordingly, in figure 5b we showed that correlation between precipitation-temperature – and thus copula parameter – may exhibit significant variation within a time frame, based on the selected sample. In other words, we wanted to point out that it is necessary to check if the observed correlation strength was purely by chance. Therefore, the conclusion that can been drawn from copula parameter – as a measure of strength of dependence – may be deceptive in certain cases.

**Autocorrelation:** 'time series are dependent on a delayed copy of themselves'?

We will change this sentence to make it clearer.

**Correlation of data with the same rank:** use the term 'ties'. I would remove subsection 3.5.1 because there is just one subsection at that hierarchy level.

Agreed.

**Selecting suitable copula families:** I would remove the NSE and RMSE part (l. 399-400).

P-value was used as the statistically sound goodness-of-fit metric in the original manuscript. As mentioned earlier, we do agree that NSE and RMSE are not traditional copula performance measures but they have been widely used in hydroclimatic applications. On the contrary to p-value that only speaks to whether the data are from a certain copula family, NSE and RMSE provide a sense of how close are empirical and copula-predicted probability levels. These metrics are not to replace p-value, but to provide further information to the user.

**Decision support framework for applying copulas:** would add 'to jointly model P and T' because some of the points are very specific (particularly the one on ties). I would re-order the different steps and put pre-treatment steps such as removal of autocorrelation (2), testing for non-stationarity (3) and ties (4) before dependence assessment (2) and copula fitting (6). I

would also include two additional steps: (x) visual inspection of dependence structure and (x) goodness-of-fit testing. My new suggested order is the following: (1) scale (if this is even important), (2) removal of autocorrelation, (3) removal of ties, (4) testing for non-stationarity, (5) visual dependence assessment, (6) computation of dependence metrics, (7) copula fitting, (8) goodness-of-fit tests. Maybe you could even have two main parts called (A) pre-treatment and (B) copula analysis.

We highly appreciate reviewer's suggestion on re-ordering the steps. We found dividing the paper in two main parts very useful. We will consider changing the structure when revising the manuscript.

**Concluding remarks:** I would discuss which parts of the decision framework are transferable to other variable pairs and which ones are specific to P-T analyses.
We will include such a discussion.

**Structure and language:** The manuscript generally has a nice flow and would profit from some editing.
Thanks, will be done.

**References:** Some additions from the statistical literature required as specified above.

We will add further statistical references to this manuscript.

**Figures:**
In general, I would recommend the use of subplot labels (a, b, c) to facilitate referencing.
Will be added where missing.

Figure 2: What do these turquoise bars on the left and lower part of the figure to the right mean?
Uniformly distributed histograms of the margin. We will mention this.

Figure 3: would remove the grey borders in the figures to the right (point clouds).
Not sure what the reviewer means here: they are x and y borders.

Figure 4: would use distinct colors in the different subplots (different shades of turquoise are used for lakes).
We will adjust this.

Figure 5: As mentioned above, I do not see the value of the analysis presented in 5b.
Through this figure, the changing dependence strength and significance can be shown, when a combination of different years are selected to form a decade and compute the correlation.

Figure 6: remove random black borders and increase legend (one should be enough).
We will adjust this.

Figure 7: Would recommend to add isolines to the scatterplots in 6b.
Fig. 6b shows autocorrelation. We are not sure how to add scatterplot to this figure.

Figure 8: Would recommend to restructure figure according to the steps order suggested above.
We will consider this option when revising and restructuring the manuscript.

**Minor points**

- I am less familiar with the term 'co-dependence' than 'interdependence'. Evtl. reword? (e.g. l. 12).

Agreed.

- 11-13. I would restructure the sentence and start with the subject 'Several multivariate analysis approaches have….. to account for precipitation….

Agreed.

- L. 55: I would talk about 'joint' instead of 'compound'.

Agreed.

- The use of commas could be improved, e.g. l. 70 'At the annual resolution,…' or l. 73: 'However,…'

We will check this.

- L. 136: 'can be' instead of 'needs to be'

Agreed.

- .139: 'provide' instead of 'provides'

Agreed.

**Comment/Response to reviewer #2: Geoff Pegrem**

What a pleasure it was to review this article. This is possibly the best Hydrometeorological paper that I have read in the last few years and is a must-read in this genre. It is targeted at authors involved with, or starting off to work with, copulas in time series. The difficulty that presents itself when analyzing time series characterized by serial correlation, is that that for analysis, modelling or forecasting, the leading question is: 'how do I get a handle on this problem?' The beauty of the paper is that it a distillation of ideas into a rubric for preparing an analysis of one or more time series, to be finished off with a flow chart for guidance.

We appreciate the reviewer for their very positive feedback and recognizing the value of this work. The motive behind writing this manuscript was to provide starters in this field with a guideline to find their way through many-existing non-statistical literature. We are very happy to see, this was perceived by the reviewer as well.

It is an important reminder and guide for time series analysis, and is not only tutorial, but is wisely, simply, and authoritatively compiled. In my judgement this should be published once some small issues have been dealt with. For example, the authors should attend to some cosmetic suggestions to fix the few spelling and grammatical errors, as well as embellishments in the figure and table captions to make them more readable. Again, stylistically, it would improve the readability if you either add a space between all paragraphs or indent the leading line. Also, I could not find 'saturation water vapor mixing ratio' in this paper. My more pertinent remarks follow.

We again appreciate the reviewer's positive comment. We will carefully revise the manuscript for potential grammatical errors and otherwise. We will change the referenced paper where we explained 'saturation water vapor mixing ratio' from (Vaghefi et al., 2014) to (Vaghefi et al. 2019). Where it is stated: 'A sequence of processes due to increasing greenhouse gasses, could be summarized as (i) increases in air temperature and its capacity to hold more water…'

In section 3, line 326, you state: 'We adopted a copula framework on July, because it has significant correlation at both daily and monthly resolutions (Fig. 5(a)-(b)).' Did you try lagging the daily precipitation and streamflow data? Surely the delay depends on the size of catchment.

We only focused on dependence structure between precipitation and temperature variables for this manuscript. Size of catchment can potentially have an impact on the dependence structure between precipitation and temperature too, and we will clarify the text.

Line 338: 'It is important to consider the degree of autocorrelation in the studied data.' Without pushing my co-authorship of a relevant paper, to check the effect of autocorrelation, you might like to look at: Sugimoto, Takayuki, András Bárdossy, Geoffrey G.S. Pegram, and Johannes Cullmann (2016), Investigation of hydrological time series using copulas for detecting catchment characteristics and anthropogenic impacts, Hydrol. Earth Syst. Sci., 20, 2705 -2720, doi:10.5194/hess-20-2705-2016.

We thank the reviewer for his suggested reference and will consider this paper.

In Figure 6: What is the spread of the confidence intervals - 95%? In (d) it looks like

100%

It is on 95% interval. We will state that in the figure.

In Figure 8: that's a very helpful flow-chart - especially the '!!'

Thanks again for recognizing the value of this chart.

In Table 2: please define the symbols in the caption to help the reader: ERMS, ENS,Sn and Dn.

We will modify. Thank you.

There are also a few minor suggestions that I have made for alteration, so I am uploading my marked-up copy of the paper with this review for the authors' information.

We appreciate the reviewer for his effort in evaluating the paper thoroughly and providing valuable comments.

Well done!

Thanks again.

---

## Author Comment (AC3) · 2 Nov 2020

We appreciate Prof., Serinaldi's effort to assess the paper critically and point out weaknesses. Most of the mentioned points have also been covered by reviewer #1, which shows the importance of raised aspects. We will strengthen the literature review that is covered in the manuscript. Further, we will certainly address all technical comments pointed out by the reviewers and commenters.

However, I (Faranak Tootoonchi) would also like to discuss this particular comment: * I also think that this type of papers should be written/supervised by people with more experience in the field; I mean names like Favre, Genest, Salvadori, De Michele, Bardossy, and some others... almost certainly, this is not a task for people with limited experience, in my opinion.*

As an avid classical music listener (and an unprofessional player), I tend to have a highly critical ear for music. Through the years, my taste in classical music have shifted from Mozart to Bach. This does not come as a surprise since Bach tends to be the most technical, accurate and demanding composer, both for listening and playing, contrary to Mozart who can be considered 'superficial' or comical. Nevertheless, for the non-professional ear, Bach is perceived as 'harsh', cacophonous and unmelodious. Professional ears tend to forget not everybody have had the same background in music. The general audience cannot fully grasp Bach, therefore this 'harshness' can be quite discouraging to them. Also, it is crucial to remember many professional players/listeners have found their way toward exquisite music through contemporary composers or composers with comparatively limited experience. It is often discouraged to start listening/playing classical music by chasing Bach, because unnecessary 'harshness' may not only be useless for the progress of music, but also kills the joy of listening to music and also is harmful for the future generation of musicians.

---

## Author Comment (AC4) · 2 Nov 2020

We appreciate that he find our paper interesting. We will revise the literature review to include related literature in this research. We also appreciate his point, as it was raised by reviewer #1 to highlight pitfalls which are more important when considering other hydroclimatic variables.

---

## Author Comment (AC5) · 2 Nov 2020

Thanks for recognizing the value of this work and taking the time to comment. We will further investigate the available literature to give readers a more concrete background on copulas both in hydrology and statistics literature. We will also consider suggested literature.